# Disentangling genetic effects on transcriptional and post-transcriptional gene regulation through integrating exon and intron expression QTLs

Anneke Brümmer [1,2,3] ✉ & Sven Bergmann [1,2,4] ✉

Expression quantitative trait loci (eQTL) studies typically consider exon expression of genes and discard intronic RNA sequencing reads despite their information on RNA metabolism. Here, we quantify genetic effects on exon and intron levels of genes and their ratio in lymphoblastoid cell lines, revealing thousands of cis-QTLs of each type. While genetic effects are often shared between cis-QTL types, 7814 (47%) are not detected as top cis-QTLs at exon levels. We show that exon levels preferentially capture genetic effects on transcriptional regulation, while exon-intron-ratios better detect those on co- and post-transcriptional processes. Considering all cis-QTL types substantially increases (by 71%) the number of colocalizing variants identified by genome-wide association studies (GWAS). It further allows dissecting the potential gene regulatory processes underlying GWAS associations, suggesting comparable contributions by transcriptional (50%) and co- and post-transcriptional regulation (46%) to complex traits. Overall, integrating intronic RNA sequencing reads in eQTL studies expands our understanding of genetic effects on gene regulatory processes.

Expression quantitative trait loci (eQTLs) are genetic variants associated with gene expression levels. While cis-eQTLs directly affect the expression of nearby genes, trans-eQTLs indirectly modulate the expression of distal genes by affecting nearby regulatory genes or elements. Mapping eQTLs has emerged as a powerful tool to identify functional genetic variants that affect gene expression and has been applied in different cell types, tissues, human populations, during ageing, upon infection, and between sexes[1–6]. eQTLs were found to colocalise with genetic variants associated with human traits through genome-wide association studies (GWAS), suggesting a causal role for gene expression in mediating such traits[7,8]. However, the extent of colocalization between GWAS variants and eQTLs was rather small (~21% of GWAS variants on average per trait[9]). The reasons could be

that sample sizes of eQTL studies are generally smaller than those of GWAS studies, which may not allow resolving eQTLs with weaker effects, that eQTLs may not have been determined in the trait-relevant cell types, or that eQTLs may not capture the trait-relevant gene regulatory processes.

Even though eQTLs provide a strong indication for a genetic variant implicated in the regulation of gene expression, the specific regulatory process affected remains ambiguous. It could range from transcription regulation, RNA splicing and processing to the regulation of RNA stability. To overcome this ambiguity, QTLs have been mapped for a variety of molecular phenotypes, such as DNA accessibility, DNA methylation, histone modifications, transcription factor binding, splicing ratios, polyadenylation site usage, ribosome-binding, and protein

[1]Department of Computational Biology, University of Lausanne, Lausanne, Switzerland. [2]Swiss Institute of Bioinformatics, Lausanne, Switzerland. [3]Bioinformatics Competence Center, University of Lausanne, Lausanne, Switzerland. [4]Department of Integrative Biomedical Sciences, University of Cape Town, Cape Town, South Africa. ✉e-mail: anneke.brummer@unil.ch; sven.bergmann@unil.ch

levels, revealing valuable insights into the genetic effects on specific gene regulatory processes[1,10–14]. While some of these molecular phenotypes can be quantified from RNA-Seq data, like gene expression for eQTLs, others require data from more advanced experimental high-throughput methods. As a consequence, such QTLs have only been studied for a few cell types or tissues and with relatively small sample sizes, limiting the statistical power for detecting QTLs with smaller effects. Thus, a way to infer the gene regulatory process affected by a genetic variant directly from gene expression measurements would be very valuable for advancing the understanding of the molecular mechanisms of eQTLs.

Over the past decade, it has become clear that intronic RNA-Seq reads contain valuable information about gene regulation. Gaidatzis et al.[15] showed that intron expression levels of genes are, like exon expression levels, primarily determined by transcription, while the ratios of exon to intron expression levels, cancelling out transcriptional influences, are more sensitive to post-transcriptional regulation. This approach has been widely applied by others to distinguish between transcriptional and post-transcriptional gene expression changes[16–18]. Another study demonstrating the value of intronic reads was by La Manno et al.[19], who used intronic and exonic RNA-Seq reads to estimate precursor and mature mRNA levels in single cells, allowing them to predict the future transcriptional regimes of individual cells from these, under the assumption that the transition from precursor to mature mRNA (i.e. RNA processing) is constant. This method (termed RNA velocity) has become a standard in single-cell RNA-Seq analyses, and modifications of it have already been proposed[20–22]. Together, analysing intronic RNA-Seq reads—which are contained in RNA-Seq data, even from polyA-selected RNA[15]—on top of exonic reads provides a deeper understanding of gene regulatory processes.

Here, we investigate the use of intronic RNA-Seq reads to improve the understanding of genetic effects on gene regulation. We determine QTLs for exon and intron expression levels of genes, as well as their ratio, using data from 901 lymphoblastoid cell lines (LCLs) from European individuals. We detect thousands of genetic variants associated with each of the three gene expression measures, including 47% that were not detected as a top cis-QTL for exon expression levels. Considering all QTL types increases the fraction of GWAS variants colocalizing with QTLs (from 18% for exon-level QTLs to 26% for all three QTL types). Furthermore, we show that integrating the information from all QTL types improves our understanding of the impact of genetic variants on gene regulatory processes and complex traits.

## Results

### Detection of QTLs for exon and intron expression levels and their ratio

To better understand the effects of genetic variants on different gene regulatory processes (Fig. 1A), we analysed QTLs for exon expression levels (referred to as exQTLs), and intron expression levels (inQTLs), and their ratio (or, equivalently, their log2-difference; referred to as ex-inQTLs). Exon and intron expression levels for each gene were quantified from RNA-Seq data obtained from lymphoblastoid cell lines (LCLs) by the CoLaus and Geuvadis consortia[2,23]. Although the number of intronic RNA-Seq reads per sample was about one order of magnitude lower than those of exonic reads, we verified that intronic gene expression levels represent biologically meaningful quantities in our data set (see Supplementary Note 1 and Supplementary Fig. 1). We then assessed that genetic associations with different gene expression measures (referred to as different QTL types below) were reproducible in two independent data sets, the CoLaus[23] and European samples of the Geuvadis data set[2], despite different sequencing depths and proportions of intronic RNA-Seq reads (see Supplementary Note 2 and Supplementary Fig. 2). To maximise the power for detecting QTLs, we combined the two data sets, resulting in a total of 901 samples. Using this combined data set, we detected significant exQTLs, inQTLs, and ex-inQTLs (FDR < 5%) for 78%, 73%, and 64% of tested genes, respectively, corresponding to 8753, 7660, and 5783 genes (Fig. 1B; Supplementary Data 1). We found that the fraction of QTLs located upstream of associated genes, where transcription regulatory regions are preferentially located, was significantly larger among exQTLs (32%) than among the other two QTL types (Fig. 1C). In contrast, the fraction of QTLs located within the transcribed gene regions, potentially affecting post-transcriptional regulation, was significantly larger for ex-inQTLs (56%).

### Genetic effects are frequently shared between QTL types

Since for many genes we found several QTL types (6904 out of 10804 genes with any QTLs), we further investigated the sharing of genetic effects between QTL types. We considered QTL effects as shared if the positions of the top QTLs, i.e. the most significantly associated variants

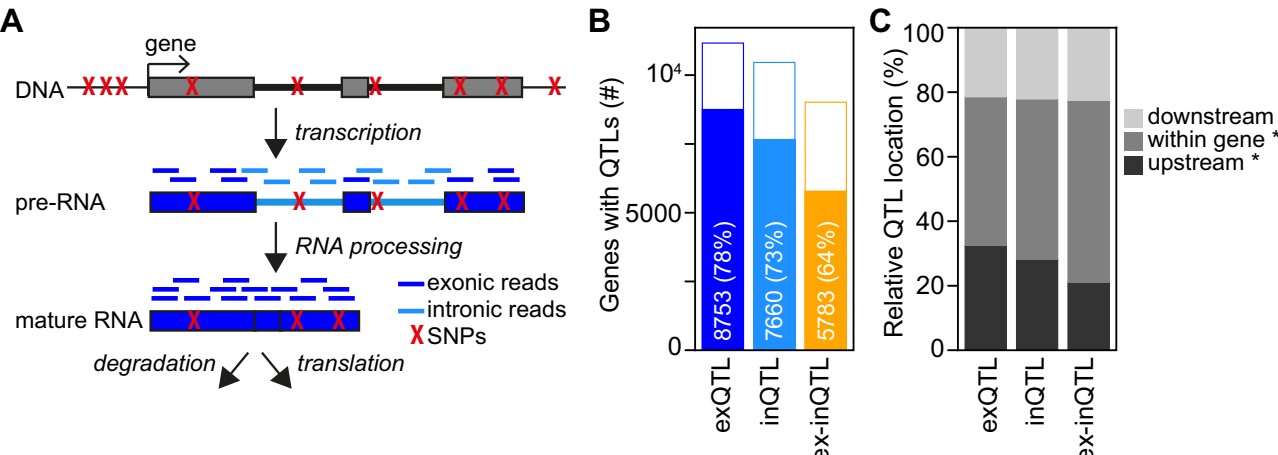

**Fig. 1 | Identification of cis-QTLs for exon and intron expression levels and their ratio. A** Schematic representation of the steps of gene expression regulation, from DNA over pre-RNA to mature RNA. Genetic variants that could potentially affect different steps are shown in red, and RNA-Seq reads mapping to exons and introns are indicated. **B** Number and percentage of genes with cis-QTLs (filled bars) of tested genes (full bars) for different QTL types. **C** Location of top cis-QTLs relative to their associated genes for different QTL types. The fractions of QTLs upstream of and within genes are each significantly different between QTL types (upstream: $p$ = 1e−9, 1e−52, and 1e−21; within: $p$ = 1e−6, 1e−34, and 1e−14 for comparisons between exQTLs and inQTLs, exQTLs and ex-inQTLs, or inQTLs with ex-inQTLs, respectively, calculated using two-sided Fisher exact test).

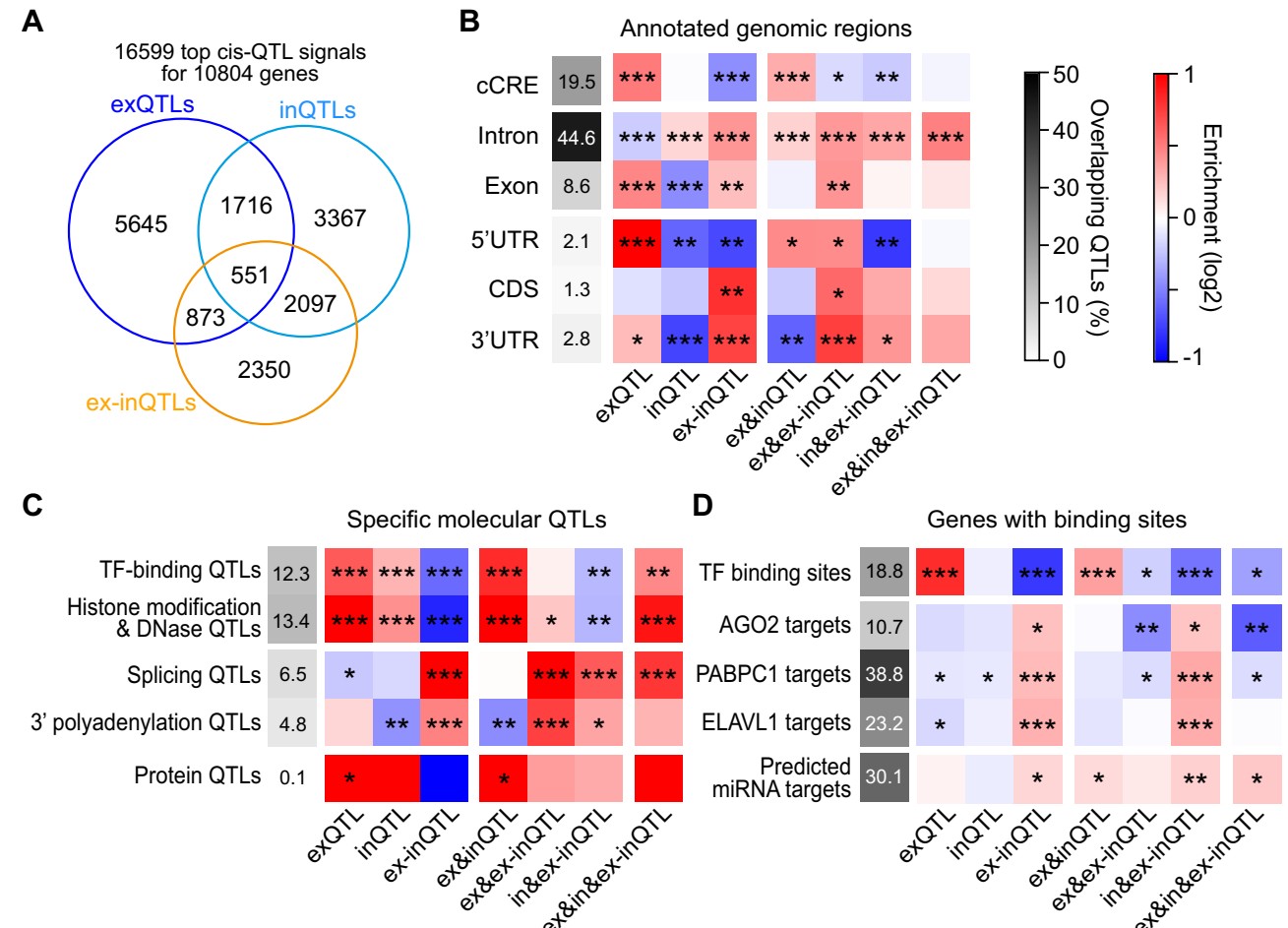

**Fig. 2 | Sharing between top cis-QTL signals, and location within specific genomic regions, overlap with other molecular QTLs, and with genes with certain binding sites. A** Venn diagram showing the sharing between different types of cis-QTLs. In total, 16,599 independent top cis-QTL signals for 10,804 genes were detected. **B** Enrichment within annotated genomic regions (represented as a blue-red heatmap) for different types of cis-QTLs and shared cis-QTLs relative to cis-QTLs not contained in or shared with that group of cis-QTLs. Asterisks indicate a significant enrichment (\*, $p < 0.05$, \*\*, $p < 0.001$, \*\*\*, $p < 0.00001$, two-sided Fisher's exact test). The overall overlap for all cis-QTLs with different genomic regions is indicated in grayscale. cCRE: candidate cis-regulatory element, UTR: untranslated region, CDS: coding sequence. **C** Similar as **B**, but for enrichment in overlap with specific molecular QTLs identified for LCLs. **D** Similar as **B**, but for enrichment within genes harbouring certain binding sites determined in LCLs, considering top cis-QTLs located within genes.

for a given gene and for each type, were identical or if both were assigned to the first conditional cis-QTL signals for both types and had consistent effect directions (see Methods). This approach revealed substantial sharing (for 5046 genes, including 2248 genes with shared QTL types with identical top QTL variants; Supplementary Data 2). Most sharing occurred between inQTLs and ex-inQTLs (referred to as in&ex-inQTLs), observed for 2648 genes, corresponding to 52% of genes with both such QTLs, followed by exQTLs and inQTLs (ex&inQTLs), observed for 2267 genes, corresponding to 39% of genes with both such QTLs (Fig. 2A). Shared exQTLs and ex-inQTLs (ex&ex-inQTLs) were slightly rarer, observed for 1424 genes, corresponding to 28% of genes with such QTLs. 551 genes presented QTL signals shared between all three QTL types (ex&in&ex-inQTLs; 12% of genes with the three QTL types). The direction of the QTL effects was mostly identical for shared ex&inQTLs, supporting a predominant effect on transcriptional regulation for these (Supplementary Fig. 3A). Overall, combining shared QTL signals, we identified 16,599 cis-QTL signals, of which 47% derived from inQTLs or ex-inQTLs and were not detected as top exQTL signals (Fig. 2A). These relative amounts of sharing between cis-QTL types were similar when using linkage disequilibrium (LD) or colocalization analysis[24] to evaluate sharing between QTL types (Supplementary Fig. 3B).

In our study, effects shared between QTL types beyond top QTLs—that is, between conditionally independent QTLs detectable after regressing out any stronger masking QTL signals—were relatively rare. Indeed, the majority of genes (76%, 69%, and 65% for exQTLs, inQTLs, and ex-inQTLs, respectively) presented only one (the top) QTL signal (Supplementary Fig. 3C; Supplementary Data 4). Nevertheless, we found a small but significant bias towards top ex-inQTL signals being more frequently shared with secondary exQTL and inQTL signals (5.4% and 8.8%, respectively) than the opposite (top exQTL or top inQTL signals shared with secondary ex-inQTL signals: 3.5% and 6.7%; $p < 0.05$, Fisher's exact test; Supplementary Fig. 3D). This may indicate that post-transcriptional effects on exon levels (detected by top ex-inQTLs) can be masked by other, stronger regulatory processes acting on exon levels, but these cases are rare.

Interestingly, we also detected shared effects between top cis-QTL signals of the same type but for different genes (Supplementary Data 2). Sharing of top exQTLs (14.8% of all genes with exQTLs) and inQTLs (12.6% of genes with inQTLs) was significantly more frequent than between ex-inQTLs (7.4% of genes with ex-inQTLs; $p < 1e-22$, Fisher's exact test). This agrees with a prevalent transcriptional co-regulation of neighbouring genes, as already reported[25], while

post-transcriptional regulation (detected by exon–intron ratios) appears more gene-specific.

## Different QTLs types and shared QTLs are enriched within genomic regions linked to distinct gene regulatory processes

To understand in more detail which gene regulatory processes are affected by different QTL types and shared QTLs, we analysed their location within specific genomic regions and sites.

As a first indication of their regulatory function, we examined the QTL location within annotated genomic regions (Fig. 2B). Compared to other QTL types, exQTLs were enriched in transcription regulatory DNA elements, i.e. candidate cis-regulatory elements (cCREs) for B lymphocytes, and in 5′ untranslated regions (UTRs) of the associated genes. In contrast, ex-inQTLs were depleted in these regions and were enriched in 3′UTRs, coding sequences (CDSs) and intronic regions. inQTLs were enriched in intronic regions but depleted in exonic regions. Shared ex&inQTLs showed enrichment in cCREs and 5′ UTRs, but not in 3′UTRs. Shared ex&ex-inQTLs were generally enriched in transcribed gene regions, particularly in 3′ UTRs. Shared in&ex-inQTLs were depleted in cCREs and 5′ UTRs and enriched in introns and 3′ UTRs. QTLs shared between all three types were enriched in introns and 3′ UTRs.

Next, we analysed the overlap with specific molecular QTLs identified previously for LCLs (Fig. 2C; see Methods). We found strong preferential overlaps for exQTLs and shared ex&inQTLs with QTLs for transcription factor (TF) binding, histone modifications and DNA accessibility. In contrast, ex-inQTLs and shared in&ex-inQTLs were depleted for overlaps with such QTLs and instead preferentially overlapped with QTLs for splicing and 3′ polyadenylation. Shared ex&ex-inQTLs and ex&in&ex-inQTLs also overlapped with splicing QTLs and additionally with histone and chromatin QTLs, suggesting a role of these shared QTLs in one or several of these processes.

Finally, we investigated the enrichment of groups of QTLs within measured TF-binding sites and experimentally identified target genes of post-transcriptionally regulatory RNA-binding proteins (RBPs), *AGO2*, *PABPC1* and *ELAVL1*, in LCLs, and predicted target genes of microRNAs (miRNAs) highly expressed in LCLs (Fig. 2D). exQTLs and shared ex&inQTLs showed strongest enrichment for TF-binding sites, while ex-inQTLs and shared in&ex-inQTLs were depleted in these regions. Instead, ex-inQTL and shared in&ex-inQTLs were enriched within genes with RBP- and miRNA-binding sites.

In summary, the enrichment analysis of QTL groups within specific genomic regions indicates that QTLs for exon expression levels are enriched for transcriptional effects, while ex-inQTLs rather represent effects on post-transcriptional processes. Shared QTLs seem to further dissect the diverse effects of QTL types on gene regulatory processes.

## TFs have stronger trans-effects on exon levels, while RBPs and miRNAs have stronger trans-effects on intron levels or exon-intron ratios

Our enrichment analysis focussed on top cis-QTLs, which have a high probability of being causal but may not always be due to strong genetic correlation between neighbouring genetic variants leading to similarly significant (sometimes indistinguishable) associations with the gene expression measurements. To further investigate how the effects of different gene regulatory processes are captured by the three QTL types in our study, we investigated trans-effects—on genes located on different chromosomes or at distances larger than 5 million base pairs —of regulatory factors (Fig. 3A). As regulatory factors we considered 724 TFs and 698 RBPs with any type of cis-QTL association in our data set, and 47 miRNAs with cis-QTLs previously identified in LCLs from European individuals[2]. We identified trans-QTLs of all types (trans-exQTL, trans-inQTL, and trans-ex-inQTL) for cis-QTLs of these regulatory factors, and detected 464 unique trans-QTL associations

between a regulatory factor and a potential target gene (219 for TFs, 193 for RBPs and 52 for miRNAs; Supplementary Data 5). While TFs had larger proportions of trans-associations with exon and intron levels (33% and 39%, respectively) than with their ratio (28%; Fig. 3B), RBPs and miRNAs had higher proportions of trans-associations with intron levels (50% and 43%, respectively) and exon-intron-ratios (30% and 37%, respectively) than with exon levels (20% each). Trans-exQTL associations were significantly more frequent for TFs than for RBPs, while trans-inQTL associations were significantly more frequent for RBPs than for TFs. Notably, the proportions of different types of trans-QTL associations for TFs and RBPs were similar when considering only cis-QTLs of any one type (Supplementary Fig. 4A), indicating that all types of cis-QTLs of regulatory factors enable similar functional trans-effects.

As sequence similarity between the cis-region around a QTL and the trans-gene can lead to read misalignments and to false positive trans-associations, we confirmed that our observations (based on uniquely mapping RNA-Seq reads) hold when considering only RNA-Seq reads overlapping genomic regions annotated with unique 36-mer mappability (see Methods; Supplementary Fig. 4B). While the observed differences in the proportions of trans-QTL association types between regulatory factors remained significant ($p < 0.05$, Fisher's exact test) when requiring a moderate read overlap, the differences did not pass the significance threshold with a stringent read selection, likely due to the strongly reduced number of RNA-Seq reads leading to less accurate gene expression quantification and fewer trans-associations.

Comparing the strengths of significant correlations between genotype and gene expression measurements across trans-QTL types, we found that cis-QTLs of TFs had generally stronger correlations with exon and intron levels of trans-genes than with the exon–intron ratio. In contrast, miRNAs had stronger correlations with exon–intron-ratios of trans-genes than with exon levels. RBPs had similarly strong correlations with all gene expression measurements of trans-genes (Fig. 3C).

Examples for trans-QTL associations with cis-regulated RBPs are between *LSM11* (involved in histone mRNA 3′-end processing) and exon or intron levels of three histone genes, between *SF3A2* (a subunit of the splicing factor complex) and intron levels of *HS2ST1*, and between *TENT5A* (implicated in mRNA stabilisation) and exon-intron ratios of *UBAC2* (Fig. 3D and Supplementary Fig. 4C). Notably, genetic trans-associations were mostly not detectable as significant correlations between the gene expression measurements of cis- and trans-regulated genes. The miRNAs with most trans-associations were *miR-550a*, *miR-3667* and *miR-4513* (7 associations each), which had 9 trans-associations with exon-intron-ratios and intron levels each, and 3 with exon levels of associated genes. Of these 5, 4, and 0 genes had predicted miRNA target sites in their 3′ UTR, respectively[26].

In summary, the trans-association analysis confirms that effects on post-transcriptional regulatory processes are better detectable at exon–intron-ratios, while transcriptional regulation is better detectable through exon levels. Intron levels appeared sensitive to both types of regulation.

## inQTLs and ex-inQTLs substantially increase the colocalization with GWAS variants

Having developed a good understanding of the regulatory processes affected by different QTL types and shared QTLs, we next investigated if combining the information from the three QTL types might aid the understanding of the functional impact of GWAS variants.

We first identified QTLs that colocalized with GWAS variants. We used the regulatory trait concordance (RTC) method[8], which tests for co-localisation between variants within the same genomic region surrounded by recombination hotspots (see Methods). Of the tested top cis-QTLs, the percentage that colocalized with GWAS variants was

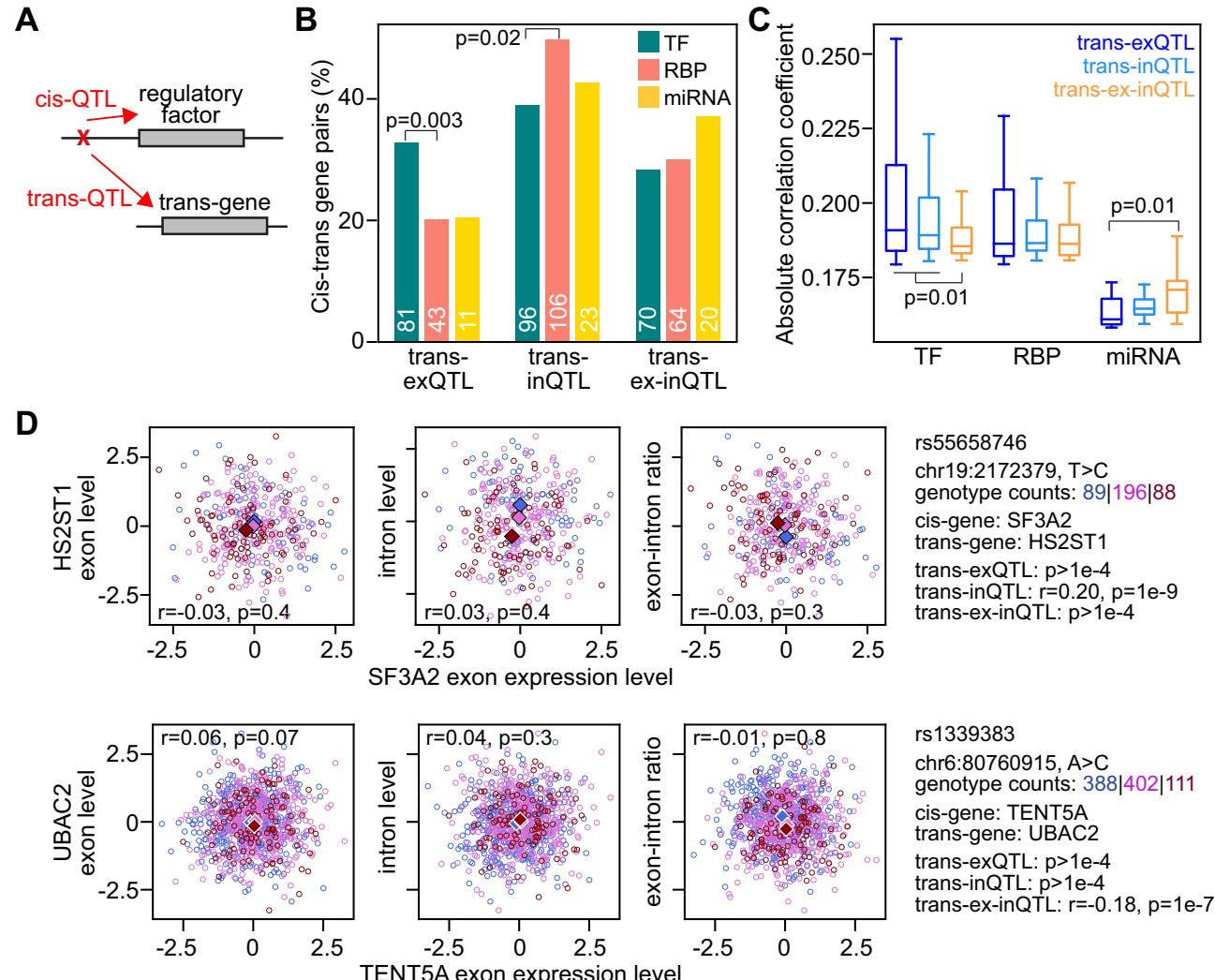

**Fig. 3 | Trans-QTL effects on exon or intron levels or their ratio for cis-QTLs of regulatory factors. A** Illustration of a cis-QTL for a regulatory factor that also has a trans-QTL effect on a (target) gene located at a distance of at least 5 Mb or on a different chromosome, which indicates a potential causal relationship between the cis-associated regulatory factor and the trans-associated gene. **B** Percentages of detected trans-QTL associations of different types for transcription factors (TF), RNA-binding proteins (RBP), and miRNAs. The numbers of trans-QTL associations are indicated at the bottom of each bar. *P* values are calculated using a two-sided Fisher's exact test. **C** Boxplots of absolute correlation coefficients (comparable to effect sizes) for the significant trans-QTL associations of different types from **B**. Boxes indicate the interquartile range of the data (second and third quartile) with a line at the median. Whiskers extend to the farthest data point lying within 1.5× the interquartile range from the box. *P* values are calculated using a two-sided Ranksum test. **D** Examples for cis-regulated RBPs, *SF3A2* and *TENT5A*, associated in trans with *HS2ST1* and *UBAC2*, respectively. Shown are scatter plots of the RBP's normalised exon levels (*x*-axis) and the normalised exon levels (left panels), intron levels (middle panels) and their ratio (right panels) of the trans-associated genes (*y*-axis). Further information on the QTL variant and the correlation coefficients and nominal *p* values of the trans-associations, determined by QTLtools[39], are indicated on the right. Pearson correlation coefficients between the expression levels/ratios of the RBPs and their trans-associated genes are indicated with *p* values inside each panel. Circles of different colours represent individuals with different genotypes, and coloured diamonds indicate the median values for individuals with that genotype.

similar for different QTL types (31–33%), supporting their similar functional relevance for human traits (Fig. 4A). Adding inQTLs and ex-inQTLs increased the number of GWAS variants colocalizing with QTLs from 3247 for exQTLs (18.4% of tested GWAS variants) to 5552 (26.3% of tested GWAS variants) for all QTL types (Fig. 4B). Thus, the number of colocalizing GWAS variants increased by 71% when considering all QTL types. As an alternative to the RTC method, we defined co-localisation based on strong LD ($r^2 > 0.8$), allowing us to evaluate almost all variants. This confirmed similar fractions of co-localising QTLs for different QTL types (20–23% of tested top cis-QTLs; Supplementary Fig. 5A), and a substantial increase in the number of co-localising GWAS variants, by 55%, when considering all QTL types (6657 or 12.5% of tested GWAS variants) instead of only exQTLs (4288 or 8.1% of tested GWAS variants; Supplementary Fig. 5B).

For 78 GWAS traits with at least 25 colocalizing variants based on RTC, colocalization increased to 27.5% of tested GWAS variants, on average per trait, (Fig. 4C and Supplementary Fig. 6A) from 16.3% for colocalization with exQTLs only. Among GWAS traits with a large amount of colocalization were skin-related (tan response) and lung function (FEV1) traits, while traits with a low colocalization included cholesterol, apolipoprotein, alkaline phosphatase, and liver protein traits. GWAS traits with a relatively large colocalization with exQTLs included testosterone, blood cell traits (eosinophil count and corpuscular haemoglobin), cognitive and neurological traits (cognitive and maths ability, intelligence, multiple sclerosis), and body fat, while GWAS traits with relatively large colocalization with inQTLs or ex-inQTLs (and low colocalization with exQTLs) included lung- and heart-related traits (FEV1 and PR interval) and glycated haemoglobin.

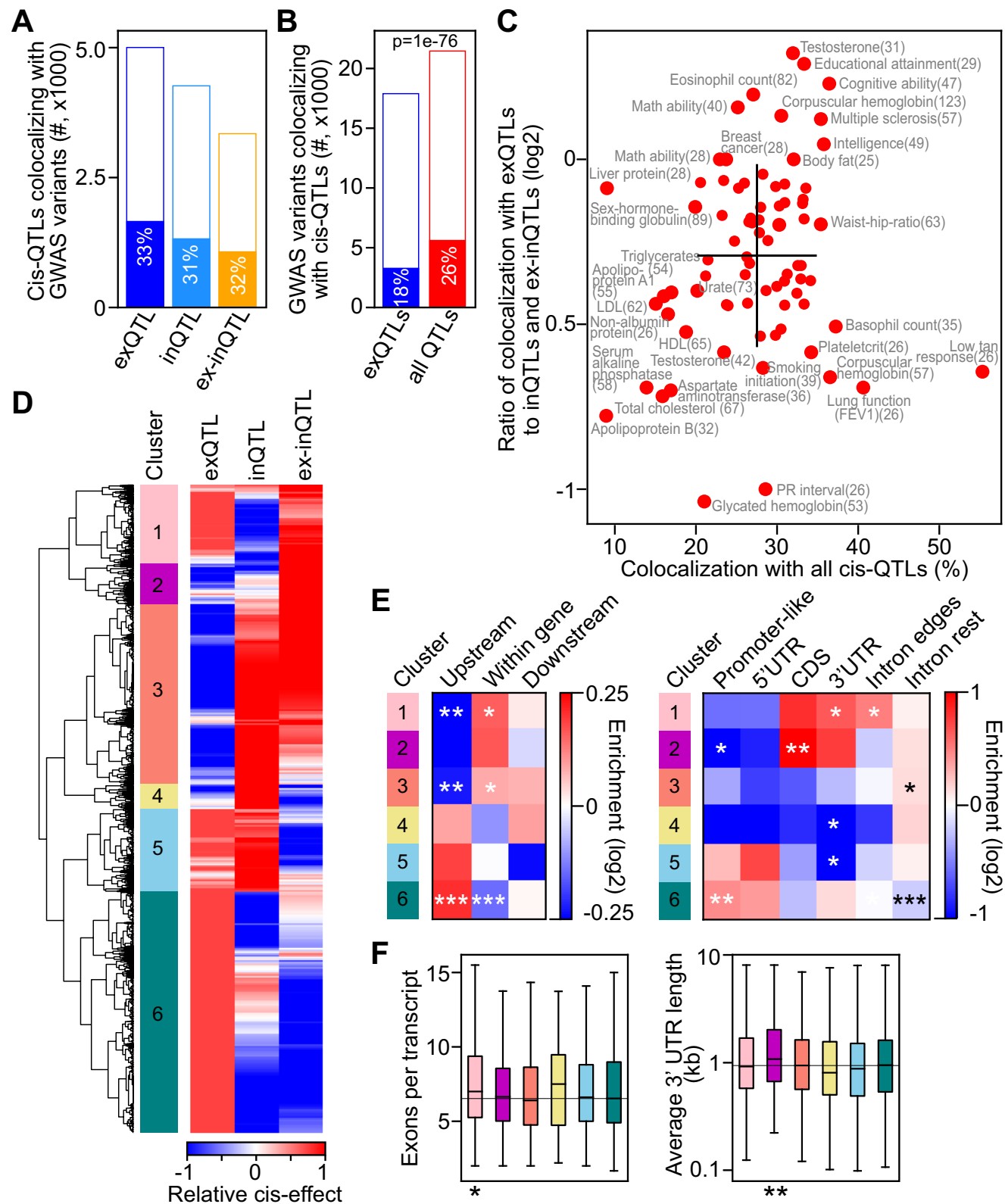

Many GWAS variants (2201 or 39.6%) colocalized with several QTL types (Supplementary Fig. 6A), potentially providing valuable information on the type of gene regulatory process contributing to a complex trait.

Overall, including inQTLs and ex-inQTLs increased the number of colocalizations with GWAS traits substantially.

**Dissecting the gene regulatory processes underlying GWAS associations**

Next, we examined if the quantified cis-effects on exon and intron levels and exon–intron ratios of QTLs colocalizing with GWAS variants might help elucidate the gene regulatory mechanisms underlying GWAS associations. We focussed on 3223 top cis-QTLs that colocalized

**Fig. 4 | Colocalization between GWAS variants and top cis-QTLs of different types. A** Number of top cis-QTLs colocalizing with GWAS variants (filled bars) of tested QTLs (full bars) for different QTL types. **B** Number of GWAS variants colocalizing with top cis-QTLs (filled bars) of tested GWAS variants (full bars) for colocalization with exQTL and all QTL types. *P* value was calculated using two-sided Fisher's exact test. **C** Percentage of colocalizing GWAS variants (*x*-axis) and log2 ratio of GWAS variants colocalizing with exQTL to those colocalizing with inQTLs or ex-inQTLs (*y*-axis), for 78 GWAS traits with at least 25 colocalizations. GWAS traits with extreme values are labelled, and the number of colocalizing GWAS variants is indicated in parenthesis. Black lines indicate the mean ± standard deviation for each axis. **D** Clustering of normalised cis-QTL effects for 3237 QTLs colocalizing with GWAS trait variants and with measured effects for all types of QTL-associations. Six distinct clusters are labelled. **E** Enrichment of QTLs upstream,

within or downstream of associated genes (left panel) and enrichment within annotated genomic regions (right panel) for cis-QTLs of each cluster compared to cis-QTLs of other clusters. Intron-edges include 100 bps at intron starts and ends. *, $p < 0.05$, **, $p < 0.01$, ***, $p < 0.0001$, calculated using two-sided Fisher's exact test. **F** Average number of exons per annotated transcript (left panel) and average 3′ UTR length (right panel) of genes associated with top cis-QTLs in different clusters (indicated by colour code from **D**). Boxes indicate the interquartile range of the data (second and third quartile) with a line at the median. Whiskers extend to the farthest data point lying within 1.5× the interquartile range from the box. *, $p = 0.011$, **, $p = 0.0014$, calculated using two-sided Ranksum test compared to genes with cis-QTLs in other clusters. The number of genes included in each box are $n = 290, 174, 671, 87, 297$ and $962$.

with GWAS variants and for which all three types of cis-effects on the target gene's expression measures were quantified, including at least one significant cis-QTL association. After normalising absolute cis-effects between QTL types and across variants, we performed hierarchical clustering with these "relative" cis-effects and obtained six clusters with distinct patterns of cis-effects (Fig. 4D). Most variants were in cluster 6, strongly affecting exon levels (37.2%), followed by cluster 3 (27.7%), affecting intron levels and exon-intron ratios, cluster 5 (12.8%), affecting exon and intron levels but not their ratios, and cluster 1 (12.2%), affecting exon levels and exon-intron ratios. Some GWAS traits colocalized with QTLs that were enriched for QTLs from a certain cluster, e.g. lymphocyte and neutrophil count traits for QTLs in cluster 2 (affecting exon–intron ratios) or haemoglobin level and metabolic biomarker traits for QTLs in cluster 6 (affecting exon levels; Supplementary Fig. 6B).

The results presented before suggest that variants in clusters 5 and 6 likely affect transcriptional processes, while variants in clusters 1, 2 and 3, with comparably strong effects on exon–intron-ratios, likely affect splicing and other post-transcriptional processes. To examine this hypothesis, we analysed the locations of the QTLs in each cluster and the structural properties of their associated genes. Indeed, variants in cluster 6 were enriched in promoter-like elements and generally upstream of genes (Fig. 4E). Variants in cluster 5 tended to be upstream and at the beginnings of genes and were depleted in 3′ UTRs. In contrast, variants in clusters 1–3 were depleted in promoters or upstream of genes, but were, in general, enriched within the transcribed gene regions. Finally, we examined the structural properties of the genes regulated by cis-QTLs in different clusters. Variants in cluster 1 were associated with genes with significantly more exons per annotated transcript, potentially indicating a more complex RNA processing for these genes, compared to genes associated with QTLs belonging to other clusters. Genes regulated by variants in cluster 2 had significantly longer 3′ UTRs, potentially indicating that these genes are under more extensive post-transcriptional regulation. Thus, the analysis of QTL location and associated genes' structural properties supports the hypothesis that variants in clusters 5 and 6 likely affect transcriptional processes, while variants in clusters 1 to 3 likely affect splicing and post-transcriptional processes.

Altogether, combining the information from different QTL types improves the understanding of the regulatory processes underlying GWAS associations.

## Examples for genetic effects on post-transcriptional and transcriptional gene regulation with relevance for complex traits

An example for a cis-QTL likely affecting co- or post-transcriptional gene regulation and colocalizing with a GWAS trait is, from cluster 1, rs2711977 (Fig. 5A), which is associated with exon levels and exon-intron-ratios of *TMEM156*, a transmembrane protein, and colocalizes with GWAS variants for monocyte count. *TMEM156* expression is also affected by an inQTL, which is not shared with the two other QTL types and did not colocalize with variants of this GWAS trait. While the

colocalizing top cis-QTL is upstream of the gene, another QTL (rs2254075), sharing the exQTL and ex-inQTL signals with the top cis-QTL, is located in the second exon, a 150 nucleotide long exon that is contained only in non-coding transcript annotations of TMEM156. Only individuals homozygous for the alternative allele express this exon, and they also exhibit lower expression of all other exons of that gene compared to individuals homozygous for the reference allele. It is possible that inclusion of this exon, which contains stop codons in all reading frames, triggers nonsense-mediated decay of that transcript, leading to its reduced overall expression. Alternatively, the two QTLs (upstream and in the second exon) together lead to simultaneous changes in splicing of that exon and the overall transcription rate. Another example is rs2278670, a cis-QTL from cluster 2 (Fig. 5B). It is associated with the exon-intron-ratio of *SMAD3*, a transcription factor functioning in the transforming growth factor-beta (TGF-β) signalling pathway, and colocalizes with GWAS variants for the lung function trait FVC. The top cis-QTLs for exon and intron levels do not colocalize with the GWAS trait variants. The top cis-ex-inQTL is located in the last exon, which contains the ~5000 nucleotides long 3′UTR. Although the top cis-ex-inQTL is not directly located within a predicted miRNA binding site, it is still possible that this QTL, or seven other cis-ex-inQTLs sharing the QTL effect and also located in the 3′UTR, interfere with miRNA targeting in their vicinity, or with binding of other post-transcriptional regulatory factors to the 3′UTR. An example for a cis-QTL from cluster 3 is rs60252802 (Fig. 5C). This cis-QTL is associated with intron level and exon-intron-ratio of *SDF4*, a calcium-binding protein involved in regulating calcium-dependent cellular activities[27]. It is located in the first intron of SDF4 and colocalizes with GWAS variants for systemic lupus erythematosus, an autoimmune disease affecting multiple organs. Inspecting the RNA-Seq read distributions of homozygous individuals with reference and alternative genotypes indicates that this cis-QTL appears to increase the probability for mis-splicing, through introducing an alternative 5′ splice site, leading to an extension of the first exon (which is part of the 5′ UTR) by almost 400 nucleotides.

Examples for cis-QTLs likely modulating transcriptional gene regulation and mediating a complex trait is rs1156242 form cluster 4 (Fig. 5D), associated with intron levels of *FAR2*, encoding a fatty acid reductase enzyme, and colocalizing with a GWAS variant for interleukin-27 levels. It is located in an intron of a downstream gene on the opposite strand, ERGIC2, whose exon and intron levels are both also affected by this cis-QTL, indicating a likely co-transcriptional regulation of these neighbouring genes. Other examples are rs914615 and rs370545 from cluster 5 (Fig. 5E), associated with exon and intron levels, respectively, of *GBAP1*, a pseudogene potentially regulating the expression of its related coding gene beta-glucosylceramidase 1 through sponging of miRNAs[28], and colocalizing with GWAS variants for mean corpuscular haemoglobin concentration, and rs72844546 from cluster 6 (Fig. 5F), associated with exon levels of *SUMO2*, a small ubiquitin-like protein modifier, and colocalizing with GWAS variants for sex hormone binding globulin levels. Both of these top cis-QTLs are

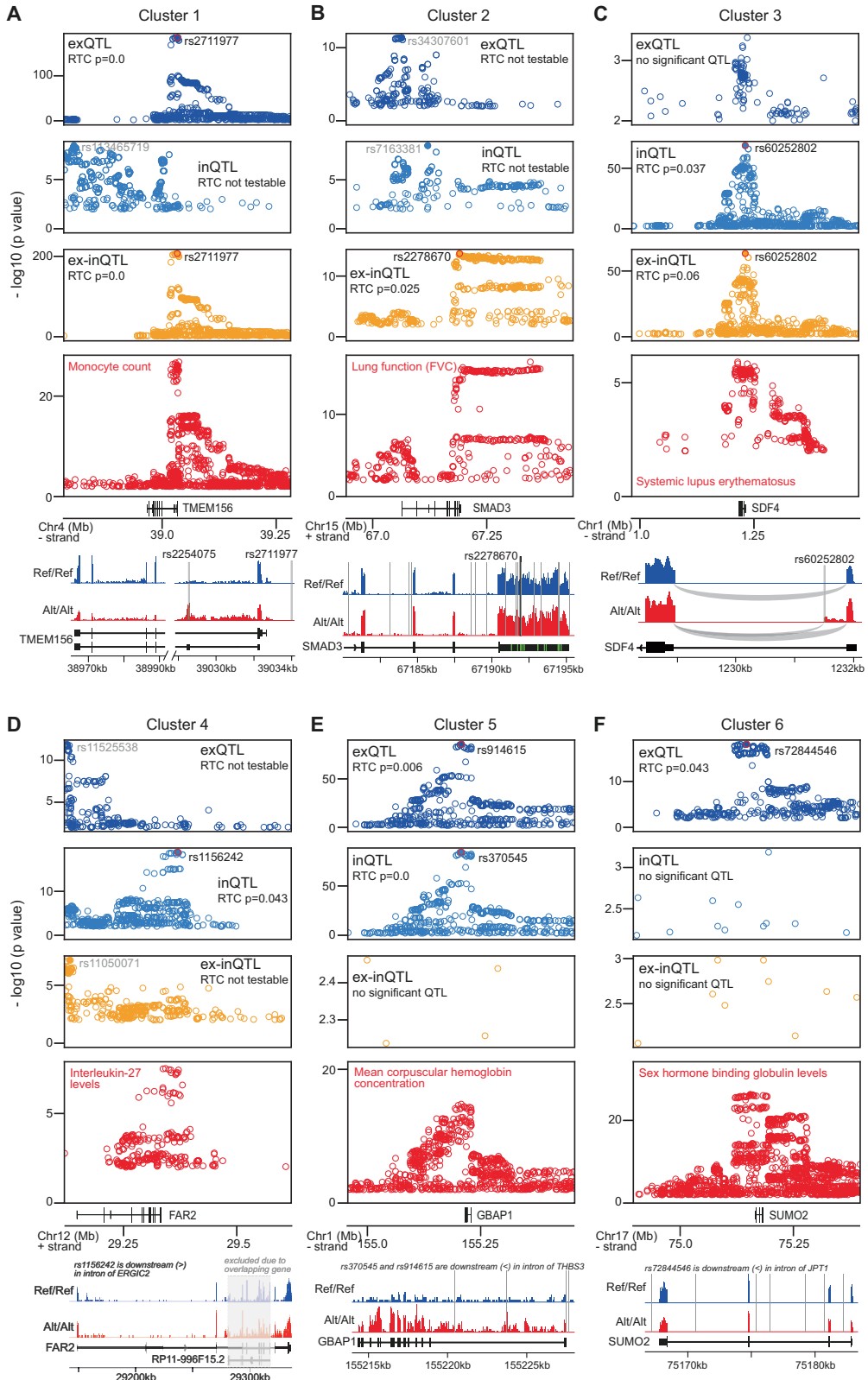

located in introns of downstream genes on the same strand, JPT1 and THBS3, respectively. A difference between these two examples is that intron levels remain unaffected for *SMAD3* (cluster 6), while they are similarly affected as exon levels for *GBAP1* (cluster 5). This could indicate that splicing is not rate-limiting for mature RNA levels for cluster 6, while the splicing rate limits the transcriptional regulation of mature RNA levels for cis-QTLs in cluster 5.

Further examples for QTLs from each cluster are shown in Supplementary Fig. 7.

## Discussion

In this study we showed that combining traditional eQTL analysis based on exon levels with the analysis of QTLs for intron levels and exon–intron-ratios increases the number of genetic variants

**Fig. 5 | Examples for cis-QTLs from different clusters colocalizing with GWAS variants.** In each subfigure, the top three panels show the -log10 nominal *p*-values (<0.01), in a region of 500 kb around the top cis-QTL (*x*-axis), for cis-QTL associations with exon levels (dark blue), intron levels (blue) and exon–intron-ratios (orange), and the fourth panel shows the −log10 *p*-values for the GWAS trait associations (red) in the same region. The rsID of the top cis-QTL(s) are indicated (in black, if colocalized with the GWAS trait variants, or in grey if not). The *p* value of the RTC (regulatory trait concordance) colocalization analysis is indicated inside each cis-QTL panel, in case colocalization was testable, i.e. QTLs and GWAS variants are within the same genomic region between recombination hotspots. The bottom panel shows examples for RNA-Seq read distributions at the associated gene from two homozygous individuals, one with reference (Ref/Ref; blue) and one with alternative (Alt/Alt; red) genotype, for the top cis-QTL variant. The positions of the top cis-QTL as well as QTLs sharing the cis-QTL signal are indicated with thick or thin lines, respectively. **A** A cis-QTL from cluster 1, associated with *TMEM156* and colocalizing with a GWAS variant for monocyte count. **B** A cis-QTL from cluster 2, associated with *SMAD3* and colocalizing with a GWAS variant for FVC, a trait related to lung function. **C** A cis-QTL from cluster 3, associated with *SDF4* and colocalizing with a GWAS variant for systemic lupus erythematosus. **D** A cis-QTL from cluster 4, associated with *FAR2* and colocalizing with a GWAS variant for interleukin-27 levels. **E** A cis-QTL from cluster 5, associated with *GBAP1* and colocalizing with a GWAS variant for mean corpuscular haemoglobin concentration. **F** A cis-QTL from cluster 6, associated with *SUMO2* and colocalizing with a GWAS variant for sex hormone binding globulin levels.

associating with gene expression measurements and expands the understanding of their effects on gene expression regulatory processes.

We found that the most significant exQTLs (which are identical to the traditionally analysed eQTLs) often affect transcriptional gene regulation, which, being a pivotal step in gene regulation, has accordingly large effects on gene expression. Smaller gene expression changes by non-transcriptional processes are less detectable by traditional eQTLs. To more comprehensively study the impact of genetic variants on gene expression, the analysis of conditionally independent eQTLs has been proposed[1,29], which are detected after stepwise regressing out from the exon expression levels stronger, masking eQTL effects. Conditional eQTLs are expected to have the ability to capture multiple independent genetic effects on gene expression, potentially including subtle effects on gene expression. Our analysis showed that conditional exQTLs do not generally correspond to the gene regulatory effects detected by inQTLs or ex-inQTLs (Supplementary Fig. 3), and thus cannot replace them. This indicates that taking into account intronic RNA-Seq reads provides information on gene expression regulation that is not accessible based on exonic reads only.

Previously, the colocalization of GWAS variants with eQTLs was found to be limited (around 21% per trait for eQTLs from 54 tissues)[9]. In our data set, the colocalization of tested GWAS variants with traditional eQTLs from LCLs was on average 16.3% per trait (for traits with 25 or more colocalizing variants). Adding inQTLs and ex-inQTLs increased the colocalization to 27.5% of tested GWAS variants per trait, thus by 69%. Integrative analysis of the effects of all QTL types suggested that 50% of colocalizing QTLs likely function in transcription regulation (QTLs in clusters 5 and 6 of Fig. 4D), while a similarly large fraction of QTLs (46.2%) had strong effects on exon-intron ratios (clusters 1, 2 and 3), indicating a likely function in post-transcriptional regulation. Thus, the contributions of transcriptional and post-transcriptional processes to complex traits appear to be similar, indicating that both types of regulation are equally functionally relevant, despite smaller effects by post-transcriptional regulation on exon levels. Indeed, QTLs with small effects on exon expression levels of genes can have important functional consequences, such as through promoting exon skipping or inclusion, or alternative splice site usage (Fig. 5 and Supplementary Fig. 7)[30].

In agreement with previous results on fold changes in exon and intron expression levels between conditions, and the difference between these changes[15], we found that the most complementary effects were between those of exQTLs and ex-inQTLs, as indicated by their enrichment in either transcription regulatory sites or regions with post-transcriptional regulatory sites (e.g. 3′ UTRs or genes with target sites of RBPs and miRNAs), respectively. ex-inQTLs were also enriched for splicing QTLs, but only 18.8% of ex-inQTLs were in strong LD with splicing QTLs, suggesting that splicing is a considerable but not the primary gene regulatory process captured by ex-inQTLs. Compared to exQTLs and ex-inQTLs, inQTLs did not show strong preferences for either of these regulatory regions, and trans-effects on intron levels

were detectable from transcription and post-transcriptional regulatory factors. To advance the understanding of potential functional roles of some introns, analysing QTLs for levels of single introns might be informative.

The sharing of top cis-QTLs indicated a sizable number of QTLs with shared effects on all three gene expression measures (Fig. 2). These QTLs had enriched overlaps with multiple specific molecular QTLs, in particular those modulating transcription regulatory processes and those affecting RNA processing, potentially indicating co-regulated effects on transcription and RNA splicing or polyadenylation, which has been described before[31].

Our approach to incorporate intronic reads identified thousands of genetic variants with effects on gene regulation. Further methodological improvements may be possible to enhance the sensitivity of detecting QTLs, such as explicitly modelling the unequal effects of polyA-selection of the sequenced RNA on exon and intron levels, or refining the gene regulatory processes affected by taking into account the QTL effect directions. To allow other researchers to use and further investigate the QTLs identified in this study with LCLs from 901 individuals of European descent, we provide them as Supplementary Data.

We have used data from LCLs in our study, and although LCLs have been widely used to study genetic effects on gene regulation in the past[2,11–13,32], their gene expression regulation may not be representative of the gene expression regulation happening in other cell-types or cells in tissues. To address the question whether our findings generalise beyond LCLs, we repeated our analysis on a small fibroblast data set (78 samples from Delaneau et al.[32]) and obtained comparable results as for LCLs (Supplementary Fig. 8). In particular, the locations relative to their associated genes for different cis-QTL types were similar to LCLs, and a sizable fraction of 42% of top cis-QTLs was not shared with top cis-exQTLs. Also, for fibroblasts, the proportions of cis-QTLs colocalizing with GWAS variants were similar for different QTL types, indicating their similar functional relevance, and the number of colocalizing GWAS variants increased considerably, by 38%, when considering all three QTL types instead of only exQTLs. This suggests that inQTLs and ex-inQTLs will enable the discovery of additional genetic effects on gene regulation more in general.

We propose the presented approach to be routinely integrated into eQTL analyses, as it is easy to implement, uses existing RNA-Seq data, and enables an expanded view of genetic effects on gene regulation.

## Methods

### Quantification of gene expression in the CoLaus and Geuvadis data sets

The CoLaus data set[23] was available in-house, while the Geuvadis data set[2] was downloaded from ArrayExpress (https://www.ebi.ac.uk/arrayexpress/experiments/E-GEUV-3/).

We aligned paired-end unstranded RNA-Seq reads from both data sets to the human genome (build GRCh38) using STAR 2.7[33] with transcript annotations from GENCODE[34] (version 34 downloaded from www.gencodegenes.org). We counted separately RNA-Seq reads that

mapped uniquely to annotated exons and introns of genes using htseq-count[35]. We considered as intronic all genic regions that were not annotated as exons in any of the gene's annotated transcripts. We shrank intron annotations additionally by 10 nucleotides at exon-intron boundaries to account for inaccuracies in splice site annotations, as done by Gaidatzis et al. [15]. As the RNA-Seq data were unstranded, we further created custom gene annotation files which contained only exonic or intronic parts of genes that did not overlap with any other annotated gene using bedtools[36]. Our exon and intron gene annotation files as well as the code to generate them are available as Supplementary Data 7. We counted as exonic the reads fully contained within the custom exon annotations (-m intersection-strict) to enrich for reads from mature RNAs. We considered as intronic all reads that overlapped our custom intron annotations (-m union), reasoning that any overlap with an intron indicates its presence. For QTL associations, we only tested genes with a median number of >10 mapped reads across individuals, whether for exonic reads (in case of exQTLs), intronic reads (in case of inQTLs) or for both (in case of ex-inQTLs). We quantified gene expression as RPKM (reads per kilobase per million reads) separately for exonic and intronic reads, using gene lengths calculated from the respective modified gene annotation files. We added to the RPKM levels a pseudo count of 0.1 and log2-transformed them. We calculated exon–intron ratios as the difference between the log2 exon and log2 intron RPKM levels.

## Mapping of cis-QTLs
We restricted our analysis to single nucleotide variants (SNVs) with a minor allele frequency >1% and a minor allele count >10. SNV positions were lifted from hg19 over to hg38 using picard-tools LiftoverVcf (https://broadinstitute.github.io/picard/). We restricted our analysis to European individuals or those with European origin, and excluded 89 samples from the Geuvadis data set that were from African individuals (from Yoruba in Nigeria; labelled YRI). For the combined data set, we merged genotypes from the CoLaus and Geuvadis data sets using vcf-merge[37].

We used the genotypes' first three principal components (PCs) as covariates and the gene expression's PCs to account for technical and other co-variabilities in the RNA-Seq data, as suggested before[38,39]. We optimised the number of PCs used as covariates by maximising the number of genes with significant (FDR < 5%) cis-QTLs, i.e., the first 70 PCs for exQTLs, the first 50 PCs for inQTLs, and the first 40 PCs for ex-inQTLs. Using that many PCs outperformed using age, sex, or the fraction of intronic reads explicitly as covariates. Notably, the number of PCs that was optimal was the same for all data sets (CoLaus, Geuvadis without YRI, and the combined data set of both). For each gene, we tested all variants located within one million nucleotides (upstream and downstream) of annotated gene starts.

We mapped cis-QTL with QTLtools 1.3[39] to obtain (1) the top cis-QTL association for each gene after empirically adjusting $p$ values using 1000 permutations, (2) all nominally significant associations, and (3) conditional cis-QTL associations after stepwise regressing out stronger, masking cis-QTL signals from the gene expression measurements.

## Sharing of cis-QTLs
We defined two cis-QTL effects as shared when the top cis-QTL positions were identical or when both top QTLs were within the first conditional QTL signal of the other with consistent effect directions (either concordant or discordant effect directions for both QTL association types). The first conditional QTL signal is composed of genetic variants that have similar, but less significant, effects on gene expression measures as the top cis-QTL (Supplementary Data 3). We used QTLtools[39] to identify the genetic variants belonging to first conditional cis-QTL signals and to identify independent conditional QTL signals that may be masked by stronger cis-QTL effects. These are

detected after stepwise regressing out the effects from stronger QTL signals from the gene expression measurements. We used the same definition of sharing to investigate sharing between conditional cis-QTL signals of ranks 1–5 (identified after stepwise regressing out up to four stronger cis-QTLs signals), between top cis-QTLs of the same type but for different genes (to investigate co-regulation of neighbouring genes), and between top cis-QTLs of the same type and for the same gene, but identified in different data sets (to investigate the reproducibility of top cis-QTL signals in the CoLaus and Geuvadis (without YRI) data sets).

We also quantified sharing between cis-QTL types based on three other approaches: (1) linkage disequilibrium (LD), requiring $r^2 > 0.8$ between QTL variants (using vcftools[37] to calculate LD between genetic variants); (2) colocalization analysis using coloc[24], considering variants assigned to the first conditional QTL signals of each type and requiring the posterior probability for colocalized association (PP4) > 0.8; (3) Storey's pi1 statistics, i.e. $q$ values[40] of the nominal $p$ values of cis-QTL association of one type for variants identified as top cis-QTLs for another cis-QTL type, and requiring $q$ value < 0.05 for both directions of sharing between the two QTL types.

## Overlap with annotated genomic regions, specific molecular QTLs and genes with regulatory sites
Genomic regions of exons, introns, coding regions, 5' UTRs, and 3' UTRs were taken from GENCODE gene annotations (version 34)[34]. We downloaded from ENCODE[41] annotations of candidate cis-regulatory elements (cCRE) identified through patterns of chromatin modifications and DNA binding factors and retained only cCREs identified for B lymphocyte cell lines. We considered the following specific molecular QTLs previously identified in LCLs: TF-binding QTLs[10], histone modification and DNase hypersensitivity QTLs[11], splicing QTLs[1], 3' polyadenylation QTLs[13], and protein level QTLs[12]. QTL positions were lifted over to hg38 using the liftOver tool and liftOver chains from UCSC. We considered the following regulatory sites experimentally determined in LCLs: TF binding clusters identified using ChIP-Seq (track: TF Clusters downloaded from UCSC Table Browser: https://genome.ucsc.edu/cgi-bin/hgTables), AGO2 binding sites identified using iCLIP-Seq[42], PABPC1 and ELAVL1 bound RNAs identified using RIP-Seq (downloaded from https://www.ncbi.nlm.nih.gov/geo/; accession codes: GSM944519 and GSM944520), plus predicted miRNA target sites from TargetScan[43] for highly expressed miRNAs in LCLs according to miRNA quantifications from ENCODE.

To compare the overlap of QTLs with the above-described genomic sites between groups of QTLs (for different QTL types and shared QTLs) we calculated for each group the enrichment as the ratio between the proportion of QTLs of that group overlapping a genomic site and the proportion of QTLs that overlap among QTLs not in that group (and not shared with them). The enrichment within genomic regions for QTLs in each cluster is calculated as above by comparing with QTLs in other clusters. For the overlap with different gene regions (Figs. 2B and 4F), only overlaps of QTLs with the associated gene's regions were considered.

## Trans-QTL associations for cis-QTLs of TFs, RBPs, and miRNAs
We determined nominal trans-associations using QTLtools[39] for cis-QTLs of transcription factors (TFs), RNA-binding proteins (RBPs) and miRNAs. We took TFs from[44] and RBPs from[45]. We tested all cis-QTLs within the first conditional QTL signal of each regulatory factor for trans-associations with exon levels (trans-exQTLs), intron levels (trans-inQTLs), and exon–intron-ratios (trans-ex-inQTLs) of genes located on different chromosomes or at distances of more than five million nucleotides from the cis-QTL. Cis-QTLs for miRNA expression levels obtained in LCLs (Geuvadis data set without African, YRI, samples) were taken from[2]. To account for multiple tests, we applied a nominal Bonferroni threshold[46]. We considered a trans-association significant if

the nominal *p*-value was smaller than 0.05/((number of tested TFs and RBPs, or miRNAs) × (number of PCs explaining 95% of the variance in gene expression between individuals)). The number of PCs was 593 for exon levels, 707 for intron levels, and 733 for exon–intron ratios. In order to compare the proportions of different trans-QTL associations across different regulatory factors, we only considered trans-associations of genes for which all three trans-QTL types were testable. We discarded trans-associations for genes that had a pseudogene or were a pseudogene of a coding gene located within one million nucleotides of the tested QTLs. Although we did not count RNA-Seq reads aligning to more than one genomic position, sequence similarities between the pseudogene and the gene in the cis-window around the QTL could still lead to mis-aligned reads.

We further evaluated a potential bias in the trans-QTL detection caused by mis-aligned reads, due to sequence similarity between the cis-region and the trans-gene, by considering only RNA-Seq read pairs that overlapped regions annotated with 36-mer mappability = 1 (downloaded from https://www.encodeproject.org/references/ENCSR821KQV/). First, we required an overlap of at least half of one of the two reads resulting in 81% of exonic and 86% of intronic reads to be considered, and second, we required a complete overlap of at least one of the two reads with a unique mappability region, resulting in 61% of exonic and 78% of intronic to be considered. Trans-QTL identification was done as described above, considering genes with a median number of >10 exonic and intronic reads across samples.

### Colocalization of cis-QTLs with GWAS variants

We downloaded GWAS associations from different studies from the GWAS catalogue (https://www.ebi.ac.uk/gwas/; version 1.0)[47]. We considered GWAS variants with an association *p*-value < 1e−8, from studies carried out with European individuals or replicated with European individuals. This resulted in 53,438 unique GWAS variants, of which 49,711 variants were also genotyped in the CoLaus/Geuvadis data set. We performed the colocalization analysis between GWAS variants and top cis-QTLs using the regulatory trait concordance (RTC) method[8] implemented in QTLtools 1.3[39], with 160 times sampling from simulated data. A colocalization was significant if the *p*-value of the RTC score was below 0.05.

As an alternative approach to the RTC method, we quantified colocalization between cis-QTLs and GWAS variants based on strong LD ($r^2 > 0.8$), considering all genotyped variants within 5 million base pairs from each other.

### Clustering of colocalizing cis-QTLs based on relative effects

To make QTL effects comparable between QTL types and across variants, we first normalised the absolute effect sizes of each variant to its maximum among the three types of cis-effects. Then, we *z*-scored the normalised absolute effect sizes across all variants separately for each of the three types of QTL effects. Hierarchical clustering of relative effect sizes was performed with scipy.cluster.hierarchy.linkage in Python3.6[48] using the Euclidean distance metrics and the weighted method for calculating distances between clusters.

### Analysis of fibroblast data from Delaneau et al. [32]

The fibroblast data was available through the Gencord data access committee and downloaded at the European Genome-Phenome Archive (www.ebi.ac.uk/ega; Study ID EGAS00001003485). Genotypes were filtered for biallelic single-nucleotide variants with minor allele frequency >1% among the 78 samples. RNA-Seq read counting from downloaded bamfiles, and gene expression quantification for fibroblasts was done exactly as for LCLs. Cis-QTL mapping was done with QTLtools using as covariates the first genotype principal component, 15 gene expression principal components for exQTLs and inQTLs, and 10 gene expression ratio principal components for

ex-inQTLs. Sharing between cis-QTL signals of different types and colocalization between cis-QTLs and GWAS variants was defined based on LD ($r^2 > 0.8$), considering variants within 5 million base pairs of each other.

### Reporting summary

Further information on research design is available in the Nature Portfolio Reporting Summary linked to this article.

## Data availability

All exQTLs, inQTLs and ex-inQTLs (associated with exon and intron expression levels of genes and their ratio, respectively) identified in this study are available as Supplementary Data. Genotypes and RNA-Seq data of the Geuvadis data set are publicly available at ArrayExpress (https://www.ebi.ac.uk/arrayexpress/experiments/E-GEUV-3/). The CoLaus data set is under controlled access and available for researchers through a data transfer agreement (contact: sven.bergmann@unil.ch). The fibroblast data is available through the Gencord data access committee at the European Genome-Phenome Archive (www.ebi.ac.uk/ega; Study ID EGAS00001003485). Gene annotations (genome build GRCh38) were downloaded from GENCODE (version 34, www.gencodegenes.org). Annotations of candidate cis-regulatory elements (cCRE) were downloaded from ENCODE. Specific molecular QTLs previously identified in LCLs were taken from the supplemental information of the respective publication for TF-binding QTLs[10], histone modification and DNase hypersensitivity QTLs[11], splicing QTLs[1], 3' polyadenylation QTLs[13], and protein level QTLs[12]. liftOver chains (from hg19 to hg38) were downloaded from UCSC. Experimentally determined regulatory sites in LCLs were TF binding clusters (track: TF Clusters downloaded from UCSC Table Browser: https://genome.ucsc.edu/cgi-bin/hgTables), AGO2 binding sites[42], PABPC1 and ELAVL1 bound RNAs (downloaded from https://www.ncbi.nlm.nih.gov/geo/; accession codes: GSM944519 and GSM944520). Predicted miRNA target sites were downloaded from TargetScan[43] (www.targetscan.org/vert_80/). miRNA expression levels in LCLs were downloaded from ENCODE. A list of TFs was taken from[44] and of RBPs from[45]. Cis-QTLs for miRNA expression levels were taken from[2]. 36-mer mappability regions were downloaded from https://www.encodeproject.org/references/ENCSR821KQV/. GWAS associations from different studies were downloaded from the GWAS catalogue (https://www.ebi.ac.uk/gwas/; version 1.0)[47].

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

## Acknowledgements

We thank Zoltan Kutalik for providing imputed genotypes for the CoLaus data set, Zoltan Kutalik and Olivier Delaneau for valuable comments, and aSciStance Ltd. for help in revising the initial manuscript. This study was supported by the Swiss National Science Foundation (Grant No. FN 310030_152724/1) to S.B.

## Author contributions

A.B. designed the project. A.B. carried out the computational analyses and prepared the results. A.B. and S.B. discussed the results. A.B. and S.B. wrote the paper.

## Competing interests

The authors declare no competing interests.
