## [Peer Review File · Nature Communications]

Disentangling genetic effects on transcriptional and post-transcriptional gene regulation through integrating exon and intron expression QTLsReviewer #1 (Remarks to the Author):

In this work, Dr. Brümmer and Dr. Bergmann explore the impact of different types of expression quantitative trait loci (eQTL) variants and their colocalization with genome-wide association study (GWAS) variants. The authors combined traditional eQTL analysis based on exon levels (exQTLs) with the analysis of eQTLs for intron levels (inQTLs) and exon-intron ratios (ex-inQTLs) to gain a more comprehensive understanding gene regulatory processes and their effect on complex traits. They showed that including inQTLs and ex-inQTLs substantially increased the number of colocalizing GWAS variants, compared to only using exQTLs. In addition, they showed that exQTLs colocalizing with GWAS trait variants capture transcriptional regulatory effects while ex-inQTLs better detect those on co- and post-transcriptional processes. Finally, they suggest that transcriptional and post-transcriptional processes contribute similarly to complex traits.

Overall, the manuscript presents a significant contribution to the field, providing valuable insights into the regulatory mechanisms underlying complex traits. The authors' rigorous analysis and clear presentation make this study a compelling read for researchers interested in the intersection of genetics, gene expression, and complex trait studies. However, the focus on LCLs limits the generalization of their findings (comment 1). In addition, their trans-eQTL mapping method might suffer from spurious associations (comment 2). Finally, better control of multiple testing is needed in some of their analyses (comment 3).

1. While the study offers valuable insights into gene regulation processes, one major limitation is the fact that the study focuses solely on lymphoblastoid cell lines (LCLs). LCLs may not fully represent the complexities of gene regulation across different contexts, i.e., tissues or cell types. Considering the context-specific nature of gene expression and regulation, it would be beneficial to explore the generalizability of the findings by incorporating data from other tissues or cell types (e.g., by analyzing publicly available data from the GTEx project). This expanded analysis would enhance the study's applicability and strengthen the conclusions regarding the broader genetic effects on transcriptional and post-transcriptional gene regulation. At minimum, the authors should acknowledge this limitation and add a discussion on the potential implications for context-specific gene regulation.

2. Sequence similarity between genes or low mappability genomic regions could lead to a large proportion of false positive trans-eQTLs. The authors should discuss the steps taken to mitigate this issue. Removing non-uniquely mapped reads, which the authors seem to do, helps reduce but not completely remove the possibility of spurious associations. I suggest that the authors rerun the trans-eQTL mapping after removing reads that are mapped to low mappability regions (e.g., those with mappability < 1.0 in the ENCODE 36 k-mer of the reference human genome) when quantifying gene expression or that they at least exclude all trans eQTL results from these low mappability regions.

3. Section "Dissecting the gene regulatory processes underlying GWAS associations": Why are you presenting / discussing results that do not pass the nominal P-value threshold for enrichment? E.g., you state that "Variants in cluster 5 were ... preferentially located upstream and at the beginnings of genes" or that "variants in clusters 3 were depleted in promoters", but those enrichment are not significant. In addition, do you adjust these enrichment p-values for the number of clusters (6) and annotations (9) tested? How many of these enrichments are significant after multiple testing adjustment?

Reviewer #2 (Remarks to the Author):

The manuscript entitled "Disentangling genetic effects on transcriptional and post-transcriptional gene regulation through integrating exon and intron expression QTLs" presents a study that integrates intronic RNA sequencing reads in QTL studies to gain a deeper understanding of genetic effects on

gene regulatory processes. The authors report thousands of cis-QTLs of each type and demonstrate that exon levels preferentially capture genetic effects on transcriptional regulation, while exon-intron-ratios better detect those on co- and post-transcriptional processes. Considering all cis-QTL types substantially increased the number of colocalizing GWAS variants and allowed dissecting the potential gene regulatory processes underlying GWAS associations, suggesting comparable contributions by transcriptional and co- and post-transcriptional regulation to complex traits. This manuscript is well-written and easy to follow. However, I have a few concerns regarding the analysis and interpretation of the results.

Major comments:

1. Many factors may influence intronic read counts. It is important to ensure that intronic read counts quantified in this study are not technical artifacts, such as contamination by genomic DNA. Further, in general, samples with less than 10 million uniquely mapped reads will be removed in the QC process of RNA-seq data. The potential bias of quantification of intron expression based on limited read counts (low depth) should be discussed, as it may lead to the lower accuracy of gene expression quantification and increased variability between samples.
2. The robustness of the ex-inQTL analysis is not clearly demonstrated. As this is a novel type of analysis, it is crucial for the authors to quantify the type I error rate of the ex-inQTL analysis to strengthen the manuscript. Doing so will provide a better understanding of the reliability of the identified ex-inQTLs and help assess the potential for false positive findings.
3. The observed concordance of exQTLs, inQTLs, and ex-inQTLs between CoLaus and Geuvadis datasets appears to be relatively low. The authors should explore potential reasons for this discrepancy, such as differences in sample size. Assessing the concordance of the same type of cis-QTLs between datasets is essential for understanding the overlap within and between cis-QTL types. If there is a low overlap of the same type of cis-QTLs between the two datasets, it may be less likely to observe a higher overlap between different types of cis-QTLs (such as between exQTLs and ex-inQTLs), even if the underlying causal variants are identical.
4. The estimation of 40.9% of top cisQTL signals derived from inQTLs or ex-inQTLs not detected by exQTLs could be biased. Although these cis-QTLs were not significant exQTLs at FDR of 0.05, it is very likely due to the difference in statistical power for different types of cis-QTLs. For example, cis-QTLs with small effects may require a larger sample size to detect. It would strengthen this manuscript to use Storey's π_1 statistic to re-estimate the overlap between different types of cis-QTLs.
5. The inference of the sharing between two types of cis-QTLs based on identical positions or similar strong associations may be either too stringent or arbitrary. Even for the same type of cis-QTLs among two different datasets, this proportion is very limited. For example, 16.7% of exQTLs have the same top QTLs between CoLaus and Geuvadis. It would be ideal to run the colocalization analysis or similar analysis between different types of QTL for the same gene and use $PP4 \geq 0.8$ as the evidence of shared signals and $PP3 \geq 0.80$ as the evidence of distinct signals. This analysis would provide a more accurate assessment of the independence between exQTLs and ex-inQTLs.

Minor comments

6. The authors claimed that ex-inQTLs are more related to co- and post-transcription. Is it possible that most ex-inQTLs are actually sQTLs, although the authors found that ex-inQTLs were enriched in sQTLs? It would be nice to know the proportion of ex-inQTLs that are also sQTLs. Further, were the ex-inQTLs enriched for splice sites? Do the genes with ex-inQTLs undergo intron retention events?
7. This study used the RTC method to test the colocalization between QTLs and GWAS. Replicating the results using other methods, such as FUSION and SMR, would strengthen the findings.

Point-by-point response to the reviewers' comments

We thank both reviewers for their thoughtful evaluation of our manuscript and for their constructive comments. We have tried to address all comments, and have added additional supplementary figures, extended the supplementary text and modified some passages in the main text.

When addressing the reviewers' comments we also carefully revisited all steps of our analysis, which led to the identification of three (minor) flaws: First, 13 out of 901 samples had been accidentally omitted due to a parsing error. Second, our implementation of our QTL sharing analysis was not exactly as described. Correcting this resulted in slightly less sharing than originally reported. Third, in Figure 3C the three types of trans-QTLs and the three types of regulatory factors had been swapped accidentally. None of these flaws impacted the results or the conclusions of our analysis, but it changed the exact values reported in the manuscript.

Reviewer #1 (Remarks to the Author):

In this work, Dr. Brümmer and Dr. Bergmann explore the impact of different types of expression quantitative trait loci (eQTL) variants and their colocalization with genome-wide association study (GWAS) variants. The authors combined traditional eQTL analysis based on exon levels (exQTLs) with the analysis of eQTLs for intron levels (inQTLs) and exon-intron ratios (ex-inQTLs) to gain a more comprehensive understanding gene regulatory processes and their effect on complex traits. They showed that including inQTLs and ex-inQTLs substantially increased the number of colocalizing GWAS variants, compared to only using exQTLs. In addition, they showed that exQTLs colocalizing with GWAS trait variants capture transcriptional regulatory effects while ex-inQTLs better detect those on co- and post-transcriptional processes. Finally, they suggest that transcriptional and post-transcriptional processes contribute similarly to complex traits.

Overall, the manuscript presents a significant contribution to the field, providing valuable insights into the regulatory mechanisms underlying complex traits. The authors' rigorous analysis and clear presentation make this study a compelling read for researchers interested in the intersection of genetics, gene expression, and complex trait studies.

We appreciate this overall positive assessment of our work.

However, the focus on LCLs limits the generalization of their findings (comment 1). In addition, their trans-eQTL mapping method might suffer from spurious associations (comment 2). Finally, better control of multiple testing is needed in some of their analyses (comment 3).

1. While the study offers valuable insights into gene regulation processes, one major limitation is the fact that the study focuses solely on lymphoblastoid cell lines (LCLs). LCLs may not fully represent the complexities of gene regulation across different contexts, i.e., tissues or cell types. Considering the context-specific nature of gene expression and regulation, it would be beneficial to explore the generalizability of the findings by incorporating data from other tissues or cell types (e.g., by analyzing publicly available data from the GTEx project). This expanded analysis would enhance the study's applicability and strengthen the conclusions regarding the broader genetic effects on transcriptional and

post-transcriptional gene regulation. At minimum, the authors should acknowledge this limitation and add a discussion on the potential implications for context-specific gene regulation.

We thank the reviewer for pointing out this potential limitation of our findings. We agree that gene expression regulation in LCLs might not be representative of the gene expression regulation in cells of human tissues. Yet, LCLs have been widely used to study genetic effects on gene expression before. One advantage of studying gene expression regulation in LCL is that they represent one specific cell type, as opposed to a mixture of cell types in tissues with diverse cell type-specific regulations occurring. However, studies of gene expression in LCLs may miss gene regulatory effects that only occur in tissues, being mixtures of multiple cell types.

We agree that it would be very interesting to explore the generalizability of our findings in the GTEx tissue data. However, we believe that conducting such an analysis is a huge endeavor, would change the focus of our manuscript considerably, and would definitely be worth a full follow-up paper. Nevertheless, to address the generalizability of our results beyond LCLs, we have run our analysis on a small fibroblast data set (78 samples from Delaneau et al. Science 2019), where we found similar amounts of sharing between different cis-QTLs types, in particular 42% of top cis-QTLs (inQTLs or ex-inQTLs) being not shared with top exQTLs. Also, similar to LCLs, for fibroblasts the overlap with GWAS variants increased considerably, by 38%, when considering all three types of QTLs instead of only exQTLs. This suggests that inQTLs and ex-inQTLs will allow for discovery of additional genetic effects on gene regulation more in general.

We have now added a paragraph to the discussion (pages 11 and 12) mentioning this potential limitation of our findings in LCLs, as well as the fact that LCLs in the past have been used extensively as a model cell line to study the variation in gene expression. We also added that in fibroblasts we observe similar key findings (shown in Supplementary Figure S8 and reproduced below), suggesting that our results are generalizable.

Figure: Cis-QTLs for exon and intron expression levels and their ratio in 78 fibroblast samples.

(A) Number and percentage of genes with cis-QTLs (filled bars) of tested genes (full bars) for different QTL types. (B) Location of top cis-QTLs relative to their associated genes for different QTL types. P values comparing the fraction of genes with different cis-QTLs upstream or within the associated gene were calculated using Fisher's exact test. (C) Venn diagram showing the sharing between different types of top cis-QTLs, defined based on strong linkage disequilibrium ($r^2 > 0.8$). In total, 1367 independent top cis-QTL signals for 1196 genes were detected. The percentage of top cis-QTL signals not shared with top cis-exQTLs is indicated in red. (D) Number and percentage of top cis-QTLs in strong linkage disequilibrium ($r^2 > 0.8$) with GWAS variants (filled bars) of tested QTLs (full bars) for different QTL types. (E) Number and percentage of GWAS variants in high linkage disequilibrium ($r^2 > 0.8$) with top cis-QTLs (filled bars) of tested GWAS variants for exQTLs (blue bar) and all QTL types (red bar). P value is calculated using Fisher's exact test. The percentage increase in GWAS variants in strong LD with cis-QTLs when considering all QTL types as opposed to only exQTLs is indicated.

2. Sequence similarity between genes or low mappability genomic regions could lead to a large proportion of false positive trans-eQTLs. The authors should discuss the steps taken to mitigate this issue. Removing non-uniquely mapped reads, which the authors seem to do, helps reduce but not completely remove the possibility of spurious associations. I suggest that the authors rerun the trans-eQTL mapping after removing reads that are mapped to low mappability regions (e.g., those with mappability < 1.0 in the ENCODE 36 k-mer of the reference human genome) when quantifying gene expression or that they at least exclude all trans eQTL results from these low mappability regions.

We agree that trans-QTL detection is sensitive to sequence similarity between cis-region and trans-gene, which can lead to false positive trans-eQTL associations. Previously, we have tried to minimize this bias, by considering only uniquely mapped reads and by excluding trans-eQTL associations, where a pseudogene of the trans-associated gene (or vice versa) was annotated in a region of 1 million base pairs around the QTL position.

To further address this potential bias, we have followed the reviewer's suggestion and performed additional trans-QTL analyses imposing more stringent constraints on RNA-Seq read mappability, which we report in Supplementary Figure S4B (reproduced below) and on page 6 of the results section. Specifically, in these analyses we only considered RNA-Seq read pairs overlapping genomic regions which are annotated with 36-mer mappability=1 (according to ENCODE). We used two approaches with different stringency for discarding reads with insufficient evidence for unique mappability: First, we only kept read pairs, where at least one of the two reads aligned with half or more of its length to such a region (removing 19.5% of exonic and 13.9% of intronic reads per sample on average). Second, we only kept read pairs, where at least one of the two reads aligned with its full length to such a region (removing 39% of exonic and 22.1% of intronic reads). In both cases, we detected similar trends in the proportions of trans-association types (with exon or intron levels or their ratio) for cis-QTLs for different regulatory factors (transcription factors, RNA-binding proteins, and miRNAs). While the differences in the proportions of trans-QTL types between regulatory factors were still significant ($p < 0.05$) in the first approach, they were not significant in the second approach, likely due to the strongly reduced number of RNA-Seq reads resulting in less accurate gene expression quantifications.

3. Section "Dissecting the gene regulatory processes underlying GWAS associations": Why are you presenting / discussing results that do not pass the nominal P-value threshold for enrichment? E.g., you state that "Variants in cluster 5 were ... preferentially located upstream and at the beginnings of genes" or that "variants in clusters 3 were depleted in promoters", but those enrichment are not significant. In addition, do you adjust these enrichment p-values for the number of clusters (6) and annotations (9) tested? How many of these enrichments are significant after multiple testing adjustment?

We agree that our discussion should focus on the nominally significant results. We have rephrased our text (page 8) accordingly.

A strict Bonferroni correction is often overly conservative when the tests are not strictly independent, which is likely in our case, and marginally significant results may still be suggestive on the trend (there is no principled reason for 1/20 being a widely accepted threshold). The nominal Bonferroni cutoffs are $0.05/54=9.3e-4$ to correct for the number of clusters and annotations, and $0.05/6=8.3e-3$ to correct for the number of clusters. To allow the reader to get a better impression of the significance levels, we decided to mark with three asterisks p-values below $1e-3$ (surviving stringent correction), with two asterisks p-values below $1e-2$ (surviving corrections for clusters), and with a single asterisk p values below 0.05 (no correction) in Figure 4E and F (reproduced below). We also used a similar encoding for p-values represented as asterisks in Figure 2B, C and D (*, $p<0.05$; **, $p<1e-3$; ***, $p<1e-5$).

Reviewer #2 (Remarks to the Author):

The manuscript entitled “Disentangling genetic effects on transcriptional and post-transcriptional gene regulation through integrating exon and intron expression QTLs” presents a study that integrates intronic RNA sequencing reads in QTL studies to gain a deeper understanding of genetic effects on gene regulatory processes. The authors report thousands of cis-QTLs of each type and demonstrate that exon levels preferentially capture genetic effects on transcriptional regulation, while exon-intron-ratios better detect those on co- and post-transcriptional processes. Considering all cis-QTL types substantially increased the number of colocalizing GWAS variants and allowed dissecting the potential gene regulatory processes underlying GWAS associations, suggesting comparable contributions by transcriptional and co- and post-transcriptional regulation to complex traits. This manuscript is well-written and easy to follow. However, I have a few concerns regarding the analysis and interpretation of the results.

We appreciate the positive comments and addressed the concerns below.

Major comments:

1. Many factors may influence intronic read counts. It is important to ensure that intronic read counts quantified in this study are not technical artifacts, such as contamination by genomic DNA. Further, in general, samples with less than 10 million uniquely mapped reads will be removed in the QC process of RNA-seq data. The potential bias of quantification of intron expression based on limited read counts (low depth) should be discussed, as it may lead to the lower accuracy of gene expression quantification and increased variability between samples.

We thank the reviewer for raising this point. As we mention in our introduction, Gaidatzis et al. (2015) have well demonstrated that intronic RNA-Seq reads from various data sets can be used to distinguish between transcriptional and post-transcriptional gene regulation, despite lower read counts than for exonic reads. We have built on this finding, and have shown that it can be extended to studying genetic effects on transcriptional and post-transcriptional gene regulation.

In our data set, the number of intronic reads per sample is 1-2 million on average (Figure S1A), which is about one order of magnitude lower than that of exonic reads. The often used QC requirement of 10 million mapped reads is usually applied on the total mapped reads (not just exonic reads) ensuring a proper RNA sequencing run. As the RNA-Seq data we used has been analyzed before and the quality of the samples was confirmed, we did not exclude any samples from our analysis.

To evaluate that intronic expression levels are not dominated by artifacts despite the low number of available intronic reads we have now added several comparisons between the gene expression levels quantified based on exonic and intronic reads (new Supplementary Figure S1, reproduced below): Briefly, while gene expression levels based on intronic reads are lower than levels based on exonic reads (as expected because intronic reads are reduced in polyA-selected RNA-Seq samples; Fig S1A), the average expression levels (across samples for each gene) based on intronic and exonic reads are well correlated across genes ($r=0.56$; Fig S1B), even among genes with very low intronic read counts ($r>0.5$; Fig S1D). Furthermore, the standard deviations of expression levels are slightly larger for intronic reads, but again well correlated with the standard deviations of gene expression based on exonic reads ($r=0.61$; Fig S1B). Only among genes with lowest intronic read counts, the correlation of standard deviations dropped considerably, below 0.3 (Fig S1D).

These results (discussed in a new Supplementary Text) show that the number of reads has an impact on the accuracy of the gene expression quantification, yet as exonic and intronic gene expression quantifications are well correlated, intronic gene expression levels also seem to represent biologically meaningful quantities, and are not dominated by artifacts.

Figure: Comparison of gene expression levels based on exonic and intronic RNA-Seq read counts.

(A) Number of RNA-Seq reads mapping uniquely to exons (blue), introns (light blue), or both (black) for samples in the CoLaus data set (top panel) and the Geuvadis, without YRI, data set (bottom panel), sorted by number of reads mapping to exons and introns. The second y-axis (right) indicates the fraction of reads mapping uniquely to introns (red). (B) Comparison of average (top panel) and standard deviation (bottom panel) of log₂ RPKM levels (for each gene across samples) quantified from exonic (x-axes) or intronic (y-axes) uniquely mapped RNA-Seq reads. Averages and standard deviation were calculated across samples for each gene (9020 genes in total). Pearson (r) and Spearman (ρ) correlation coefficients are indicated. (C) Comparison of mean and standard deviation of average log₂ RPKM levels (left panel) and their standard deviations (right panel) quantified from exonic (blue) or intronic (lightblue) reads for 10 groups of genes with decreasing numbers of intronic reads. Each group comprises 900 genes. (D) Pearson (red) and Spearman (grey) correlation coefficients between exonic and intronic average log₂ RPKM levels (left panel) and their standard deviations (right panel) for the 10 groups of genes with decreasing numbers of intronic reads.

2. The robustness of the ex-inQTL analysis is not clearly demonstrated. As this is a novel type of analysis, it is crucial for the authors to quantify the type I error rate of the ex-inQTL analysis to strengthen the manuscript. Doing so will provide a better understanding of the reliability of the identified ex-inQTLs and help assess the potential for false positive findings.

We have used the same approach for identifying ex-inQTLs as for exQTLs, using the difference between the log2 exon and intron expression levels for each gene as a phenotype instead of the log2 gene expression level based on exonic reads in case of exQTLs. Thus, the type I error (or false positive rate) should not be different from those of standard exQTL analysis. Indeed, inspecting the QQ plot (shown below) of the nominal p values for cis-QTL association tests indicates that the p-value distribution of ex-inQTLs (and inQTLs) is as well-calibrated as that of exQTLs, indicating that ex-inQTL association p-values are unlikely to be inflated.

To quantify the absolute false positive rates of our cis-QTL analysis, we would need to have a gold-standard of true and false associations, which is not available. Yet, we can estimate false positive rates for different QTL types considering the replication in the CoLaus and Geuvadis data sets and evaluate if they are different between QTL types. (For a detailed description of the replication analysis see our answer to the next comment.) Assuming that the cis-QTL associations obtained with the larger data set (CoLaus) are true, the false positive rates ($FPR = FP / (FP + TN)$) of cis-QTL associations in the Geuvadis data set (with sample size 71% of the CoLaus data set) are: for exQTLs: $FPR = 18.2\%$ ($FP = 518$, $TN = 2323$); for inQTLs: $FPR = 14.7\%$ ($FP = 439$, $TN = 2554$); and for ex-inQTLs: $FPR = 15.3\%$ ($FP = 542$, $TN = 2993$). This indicates that the false positive rate is not larger for ex-inQTLs and inQTLs than for exQTLs. (Of course the absolute FPR values are depending on the specific sample sizes of two data sets.)

3. The observed concordance of exQTLs, inQTLs, and ex-inQTLs between CoLaus and Geuvadis datasets appears to be relatively low. The authors should explore potential reasons for this discrepancy, such as differences in sample size. Assessing the concordance of the same type of cis-QTLs between datasets is essential for understanding the overlap within and between cis-QTL types. If there is a low overlap of the same type of cis-QTLs between the two datasets, it may be less likely to observe a higher overlap between different types of cis-QTLs (such as between exQTLs and ex-inQTLs), even if the underlying causal variants are identical.

We thank the reviewer for raising this point. We have improved the analysis of cis-QTL replication between the CoLaus in Geuvadis data sets in three ways (shown in Supplementary Figure S2, and reproduced below, and discussed in a modified Supplementary Text): First, we have adjusted the set of analyzed SNPs to those that were genotyped in both data sets (~6.8M SNPs). Second, we have evaluated the replication in both directions, replicating CoLaus (n=528) cis-QTLs in Geuvadis (n=373), or replicating Geuvadis cis-QTLs in CoLaus data. Third, in addition to evaluating the cis-QTL replication for all genes, we have separately evaluated this replication for genes with high or low read counts (splitting genes into two equally sized groups based on the average rank of their exonic and intronic read counts).

We found that 88-93% of genes with cis-QTLs could be replicated in a larger data set (replication of Geuvadis cis-QTLs in CoLaus), while only 61-76% were replicated in a smaller data set (replication of CoLaus cis-QTLs in Geuvadis). While this replication of genes with cis-QTLs was hardly affected by the average read count of the genes, the overall fraction of genes with cis-QTLs was higher among genes with high read count (65-77% for CoLaus and 62-65% for Geuvadis) than among genes with low read count (47-69% for CoLaus and 34-57% for Geuvadis). The percentage of cis-regulated genes with identical or shared cis-QTLs in both data sets (63-66%) was also not affected by the average read count of the genes. The fraction of genes with cis-ex-inQTLs (and cis-inQTLs) was lower than the fraction with cis-exQTLs, but the percentage of identical or shared cis-ex-inQTLs was similar for all cis-QTL types.

Thus, with this adjusted evaluation of replication, it becomes clear that all types of cis-QTLs can be reasonably replicated. Also, the effects of sample size and gene read counts on the detection and replication of cis-QTLs are better understandable now.

1. Replication of CoLaus QTLs in Geuvadis data set

		Tested in both	Significant in CoLaus	%	Significant in both	%	Same top QTL in both	Shared top QTL signal	%
All genes	exQTLs	10847	7876	73%	5952	76%	946	2955	66%
	inQTLs	9887	6698	68%	4232	63%	574	2107	63%
	ex-inQTLs	8619	4831	56%	2979	61%	377	1558	65%
Genes with high read count	exQTLs	5423	4143	76%	3105	74%	467	1569	66%
	inQTLs	4943	3825	77%	2442	64%	315	1185	61%
	ex-inQTLs	4309	2789	65%	1726	62%	198	888	63%
Genes with low read count	exQTLs	5423	3727	69%	2843	76%	479	1383	65%
	inQTLs	4943	2870	58%	1789	62%	259	921	66%
	ex-inQTLs	4309	2040	47%	1242	61%	178	669	68%

2. Replication of Geuvadis QTLs in CoLaus data set

		Tested in both	Significant in Geuvadis	%	Significant in both	%	Same top QTL in both	Shared top QTL signal	%
All genes	exQTLs	10847	6679	62%	5952	93%	946	2955	66%
	inQTLs	9887	4618	47%	4232	93%	574	2107	63%
	ex-inQTLs	8619	3437	40%	2979	88%	377	1558	65%
Genes with high read count	exQTLs	5423	3306	65%	3105	94%	467	1569	66%
	inQTLs	4943	2584	61%	2442	95%	315	1185	61%
	ex-inQTLs	4309	1914	62%	1726	90%	198	888	63%
Genes with low read count	exQTLs	5423	3081	57%	2843	92%	479	1383	65%
	inQTLs	4943	1970	40%	1789	91%	259	921	66%
	ex-inQTLs	4309	1462	34%	1242	85%	178	669	68%

4. The estimation of 40.9% of top cisQTL signals derived from inQTLs or ex-inQTLs not detected by exQTLs could be biased. Although these cis-QTLs were not significant exQTLs at FDR of 0.05, it is very likely due to the difference in statistical power for different types of cis-QTLs. For example, cis-QTLs with small effects may require a larger sample size to detect. It would strengthen this manuscript to use Storey's π_1 statistic to re-estimate the overlap between different types of cis-QTLs.

As suggested by the reviewer (this and next comment), we have now used, in addition to our definition of sharing based on first conditional cis-QTL signals, Storey's π_1 statistics, colocalization analysis (coloc; Giambartolomei et al. 2014), and linkage disequilibrium (LD) to quantify sharing between cis-QTL types. The results are shown in a new Supplementary Figure S3B (reproduced below) and discussed in the results section (page 4). Our sharing definition based on first conditional cis-QTL signals leads to very similar amounts of sharing as for coloc (requiring posterior probability for colocalized association $PP_4 > 0.8$) or LD (requiring $r^2 > 0.8$). Storey's π_1 statistics, i.e. q value (<https://github.com/StoreyLab/qvalue>), is less stringent, as it just evaluates whether a top cis-QTL for one gene expression measure also has an adjusted nominally significant association (q-value < 0.05) with the other gene expression measure. This leads to much higher amounts of sharing based on q-value, and fewer independent cis-QTL signals.

5. The inference of the sharing between two types of cis-QTLs based on identical positions or similar strong associations may be either too stringent or arbitrary. Even for the same type of cis-QTLs among two different datasets, this proportion is very limited. For example, 16.7% of exQTLs have the same top QTLs between CoLaus and Geuvadis. It would be ideal to run the colocalization analysis or similar analysis between different types of QTL for the same gene and use $PP4 \geq 0.8$ as the evidence of shared signals and $PP3 \geq 0.80$ as the evidence of distinct signals. This analysis would provide a more accurate assessment of the independence between exQTLs and ex-inQTLs.

Please see our response to the previous comment. Apart from that, we recognized that the description of our sharing definition could have been misleading, and we have refined the description (page 4). We have also adjusted the statements in the introduction and abstract “xx% were not detectable at exon levels” (to “xx% were not detected as top cis-QTLs at exon levels”), which was not completely correct, as top cis-inQTLs or top cis-ex-inQTLs may sometimes have nominal significance at exon levels, but they were not identified as top cis-exQTL signals (and also very rarely as a secondary or lower conditional cis-QTL signals, as shown in Figure S3C and D).

We also examined the evidence for distinct signals indicated by coloc’s posterior probability for independent associations ($PP3$) >0.8 , as suggested by the reviewer. Of the 4348 genes that had top cis-QTLs of all three types, 772 (17.8%), 295 (6.8%) and 467 (10.6%) had an independent top cis-exQTL, cis-inQTL, and cis-ex-inQTL, respectively. 541 (12.4%) had a shared in&ex-inQTL that was independent of the top cis-exQTL signal, 271 (6.2%) had a shared ex&inQTL that was independent of the top cis-ex-inQTL signal, and 119 (2.7%) had a shared ex&ex-inQTL that was independent of the top cis-inQTL signal. Thus, exon levels detected most independent signals (17.8%) followed by exon-intron ratios (10.4%). Signals detected by intron levels and exon-intron ratios together were often independent of exon levels (12.4%), while signals detected by exon levels and exon-intron ratios were rarely independent of intron levels (2.7%). This confirms the previously observed complementarity of exQTLs and ex-inQTLs.

Minor comments:

6. The authors claimed that ex-inQTLs are more related to co- and post-transcription. Is it possible that most ex-inQTLs are actually sQTLs, although the authors found that ex-inQTLs were enriched in sQTLs? It would be nice to know the proportion of ex-inQTLs that are also sQTLs. Further, were the ex-inQTLs enriched for splice sites? Do the genes with ex-inQTLs undergo intron retention events?

ex-inQTLs indeed showed a strong enrichment in splicing QTLs (Figure 2C), which may suggest that a large proportion of ex-inQTLs function in splicing regulation. We have now checked the overlap of our top cis-QTL types with splicing QTLs (identified in ~150 LCL samples from GTEx, p value $< 2.5e-4$). QTL positions and associated genes were identical for 5.9%, 6.1%, and 9.5% of cis-exQTLs, cis-inQTLs, and cis-ex-inQTLs, respectively, and QTL positions were in LD ($r^2 > 0.8$) and associated with the same genes for an additional 6.3%, 6.2% and 9.4% of top cis-QTLs. This adds up to a total overlap of 12.2%, 12.3%, and 18.8% for top cis-exQTLs, cis-inQTLs, and cis-ex-inQTLs, respectively. Thus, the overlap between splicing QTLs and cis-ex-inQTLs is considerably larger than for other QTL types, but it is still less than 20%, so the majority of cis-ex-inQTLs is likely not related to splicing.

The closest distance to exon-intron boundaries was significantly smaller for cis-ex-inQTLs than for cis-exQTLs or cis-inQTLs ($p < 1e-7$), but the median distance was still large (2077 nucleotides; see Figure below).

Since these two analyses indicate that cis-ex-inQTLs are not primarily related to splicing, we did not investigate further the overlap with genes undergoing intron retention.

We have added the information that ex-inQTLs are enriched for, but not primarily, splicing QTLs to the discussion section (page 11).

7. This study used the RTC method to test the colocalization between QTLs and GWAS. Replicating the results using other methods, such as FUSION and SMR, would strengthen the findings.

We thank the reviewer for this suggestion. As FUSION, SMR and also coloc all require the full summary statistics of GWAS associations, which is not available for all analyzed GWAS traits, we decided to use strong LD ($r^2 > 0.8$) as an alternative approach to evaluate colocalization between a top cis-QTL and a GWAS variant (results are shown in a new Supplementary Figure S5, reproduced below, and mentioned in the results section on page 7). This allowed us to test almost all top cis-QTLs and GWAS variants, not only those located in the same genomic region surrounded by recombination hotspots as done by the RTC method. Using this approach, we detected slightly more, but lower proportions of tested top cis-QTLs colocalizing with GWAS variants (23%, 20% and 21% for cis-exQTLs, cis-inQTLs, and cis-ex-inQTLs, respectively, as compared to 33%, 31%, 32% with RTC), and more GWAS variants colocalizing with top cis-QTLs (4288 with top cis-exQTLs and 6657 with any top cis-QTLs). Overall, the fraction of tested GWAS variants that colocalized with top cis-QTLs increased by 55% when considering all QTL types (8.1% for top cis-exQTL and 12.5% for all top cis-QTLs).

Thus, defining colocalization based on LD confirmed the results obtained by the RTC method, in particular, similar proportions of colocalization for different cis-QTL types (20-23%) and a substantial increase in the number of colocalizing GWAS variants when considering all cis-QTL types, both indicating a similar functional relevance of all QTL types.

Reviewer #2 (Remarks to the Author):

I appreciate the authors' efforts to address all my concerns.

I have just one minor comment. The authors presented a way for quantifying the false positive rates across all QTL categories and demonstrated that ex-inQTLs do not yield a higher rate than exQTLs. This serves as a strong evidence of the reliability of ex-inQTLs. Additionally, simulations could be employed as an alternative approach for measuring type I error rates.

Reviewer #3 (Remarks to the Author):

The authors have adequately addressed the questions posed by reviewer #1. The authors spent serious efforts on the major comments with regards to the generalizability across cell-types and to the robustness of the trans-eQTL analyses:

- The authors have now studied fibroblasts as well and get similar results to the previously presented results on LCLs.

- The authors now have used additional filtering on the trans-eQTLs and use two different strategies to conduct additional filtering. When using one of these filtering strategies the differences in the proportions of trans-QTL types between regulatory factors were still significant ($p < 0.05$), whereas using the second strategy this does not anymore. The authors suggest this might be due to the fact that many sequence reads had to be filtered out and that as a consequence the signal-to-noise ratio decreased, leading to a drop of significance. I find this explanation reasonable.

Point-by-point response to the reviewers' comments

Reviewer #2 (Remarks to the Author):

I appreciate the authors' efforts to address all my concerns.

I have just one minor comment. The authors presented a way for quantifying the false positive rates across all QTL categories and demonstrated that ex-inQTLs do not yield a higher rate than exQTLs. This serves as a strong evidence of the reliability of ex-inQTLs. Additionally, simulations could be employed as an alternative approach for measuring type I error rates.

We thank the reviewer for valuing our efforts in addressing his previous comments and for approving our responses.

We agree that a similar false positive rate for all types of cis-QTLs is a strong indication for the reliability of ex-inQTLs and inQTLs. Encouraged by the reviewer's comment we have decided to add this result to Supplementary Note 2: "We also used the CoLaus and Geuvadis data sets to estimate the false positive rate (FPR) or type I error of different QTL types assuming that cis-QTL associations detected with the larger data set (CoLaus) are true. In this way, in the Geuvadis data set the FPRs ($= FP / [FP + TN]$, with FP the number of false positive associations and TN the number of true negative associations) were 18.2% for exQTLs, 14.7% for inQTLs, and 15.3% for ex-inQTLs. Thus, the FPRs are similar, and in particular not larger for ex-inQTLs and inQTLs than for exQTLs, indicating the reliability of ex-inQTLs and inQTLs proposed in this study. (Notably, the absolute FPRs depend on the specific sample sizes of two data sets.)"

We thank the reviewer for mentioning that simulations could provide an additional means for measuring the type I error rate. Since our way of quantifying the false positive rates of cis-QTL associations already provided a strong indication for the reliability of also ex-inQTLs and inQTLs, we decided to refrain from conducting additional, potentially time-consuming, simulations for estimating the type I error rates.

Reviewer #3 (Remarks to the Author):

The authors have adequately addressed the questions posed by reviewer #1. The authors spent serious efforts on the major comments with regards to the generalizability across cell-types and to the robustness of the trans-eQTL analyses:

- The authors have now studied fibroblasts as well and get similar results to the previously presented results on LCLs.
- The authors now have used additional filtering on the trans-eQTLs and use two different strategies to conduct additional filtering. When using one of these filtering strategies the differences in the proportions of trans-QTL types between regulatory factors were still significant ($p < 0.05$), whereas using the second strategy this does not anymore. The authors suggest this might be due to the fact that many sequence reads had to be filtered out and that as a consequence the signal-to-noise ratio decreased, leading to a drop of significance. I find this explanation reasonable.

We very much thank the reviewer for evaluating and approving our responses to comments posed by reviewer #1.